# MicroRNA-eQTLs in the developing human neocortex link miR-4707-3p expression to brain size

**Michael J Lafferty[1,2], Nil Aygün[1,2], Niyanta K Patel[1,2], Oleh Krupa[1,2], Dan Liang[1,2], Justin M Wolter[1,2,3,4], Daniel H Geschwind[5,6,7,8], Luis de la Torre-Ubieta[8], Jason L Stein[1,2,4]\***

[1]Department of Genetics, University of North Carolina at Chapel Hill, Chapel Hill, United States; [2]UNC Neuroscience Center, University of North Carolina at Chapel Hill, Chapel Hill, United States; [3]Department of Cell Biology and Physiology, The University of North Carolina at Chapel Hill, Chapel Hill, United States; [4]Carolina Institute for Developmental Disabilities, The University of North Carolina at Chapel Hill, Chapel Hill, United States; [5]Neurogenetics Program, Department of Neurology, David Geffen School of Medicine, University of California, Los Angeles, Los Angeles, United States; [6]Center for Autism Research and Treatment, Semel Institute, David Geffen School of Medicine, University of California, Los Angeles, Los Angeles, United States; [7]Department of Human Genetics, David Geffen School of Medicine, University of California, Los Angeles, Los Angeles, United States; [8]Department of Psychiatry and Biobehavioral Sciences, Semel Institute, David Geffen School of Medicine, University of California, Los Angeles, Los Angeles, United States

*For correspondence:
jason_stein@med.unc.edu

**Abstract** Expression quantitative trait loci (eQTL) data have proven important for linking non-coding loci to protein-coding genes. But eQTL studies rarely measure microRNAs (miRNAs), small non-coding RNAs known to play a role in human brain development and neurogenesis. Here, we performed small-RNA sequencing across 212 mid-gestation human neocortical tissue samples, measured 907 expressed miRNAs, discovering 111 of which were novel, and identified 85 local-miRNA-eQTLs. Colocalization of miRNA-eQTLs with GWAS summary statistics yielded one robust colocalization of miR-4707–3p expression with educational attainment and brain size phenotypes, where the miRNA expression increasing allele was associated with decreased brain size. Exogenous expression of miR-4707–3p in primary human neural progenitor cells decreased expression of predicted targets and increased cell proliferation, indicating miR-4707–3p modulates progenitor gene regulation and cell fate decisions. Integrating miRNA-eQTLs with existing GWAS yielded evidence of a miRNA that may influence human brain size and function via modulation of neocortical brain development.

## Editor's evaluation

Lafferty et al. study the regulation of microRNA (miRNA) levels in mid-gestation human neocortical tissues as a potential contributor to brain-related phenotypes. They performed miRNA expression profiling and correlated expression quantitative trait loci (eQTLs) that locally regulate miRNA genes, searching for potential overlap. They report colocalization at SNP rs4981455 which is an eQTL for miR-4707-3p and is also associated with global cortical surface area (GSA) and educational attainment phenotypes in GWAS, and demonstrate exogenously increased expression of miR-4707-3p in primary human neural progenitor cells increases the rate of proliferation. The reported results are

interesting and important, particularly for the understanding of miRNA biology, and suggest long-suspected roles for miRNAs in neurogenesis and human brain disease.

## Introduction

Genome-wide association studies (GWAS) have identified many genetic loci influencing human behavior, cognition, and brain structure (*Grasby et al., 2020*; *Lee et al., 2018*; *Davies et al., 2018*; *Knol et al., 2020*; *Okbay et al., 2016*). Expression quantitative trait loci (eQTL) data is often used to link non-coding brain-trait associated loci with genes that putatively mediate their effects (*Zeng et al., 2022*; *PsychENCODE Consortium et al., 2015*; *Aygün et al., 2021*). Brain eQTL studies are most often conducted in bulk adult postmortem tissue and are focused on measuring mRNA expression levels from protein-coding genes. Although these methods have been successful in linking a subset of non-coding brain-trait associated loci to genes, there may be multiple mechanisms by which a single locus influences a complex trait and many loci are still unlinked to genes (*Walker et al., 2019*; *O'Brien et al., 2018*; *Visscher and Goddard, 2019*; *Watanabe et al., 2019*; *Uffelmann and Posthuma, 2021*; *Umans et al., 2021*). This suggests other types of RNAs, unmeasured in previous eQTL studies, may be mediating the genetic associations.

MicroRNAs (miRNAs) are poorly measured in standard eQTL studies because the library preparation methods effectively remove small RNAs. Library preparation methods have been developed that specifically quantify the expression of small RNAs and allow measurement of miRNA expression in large sample sizes necessary for miRNA-eQTL studies. To date, relatively few miRNA-eQTL studies have been published, and those that have, are often underpowered or in tissues not directly implicated in brain-related phenotypes (*Huan et al., 2015*; *Li et al., 2020*; *Borel et al., 2011*; *Gamazon et al., 2012*; *Civelek et al., 2013*). Given evidence that miRNAs are strongly involved in fate decisions during neurogenesis and brain development, there is an increasing need to understand the genetic basis by which miRNAs are regulated (*Nowakowski et al., 2018*; *Nowakowski et al., 2011*; *Fineberg et al., 2009*; *Volvert et al., 2012*).

Previous studies have found enrichment of brain structure and cognition GWAS heritability within regulatory elements active during mid-fetal development (*Grasby et al., 2020*; *de la Torre-Ubieta et al., 2018*; *Liang et al., 2021*). Mapping mRNA-eQTLs in human mid-gestation cortical tissue or neural progenitor cells derived from that tissue has revealed novel developmentally specific colocalizations with brain structure and cognitive traits (*Aygün et al., 2021*; *Walker et al., 2019*). These findings are consistent with the radial unit hypothesis, which posits that increases in size of the neural progenitor pool, present only in mid-gestation, leads to increases in the size of the cortex (*Rakic, 1988*; *Geschwind and Rakic, 2013*). The discovery of miRNA-eQTLs during prenatal cortical development may highlight additional molecular mechanisms by which non-coding loci influence brain-related traits.

In this study, we performed a local-miRNA-eQTL analysis in 212 mid-gestation human cortical tissue donors to discover the common genetic variation associated with expression of nearby miRNAs (*Figure 1A*). We identified 85 local-miRNA-eQTLs (variant - miRNA pairs) associated with expression of 70 miRNAs. These miRNAs were often found within host mRNAs (49 of 70 miRNAs), but the genetic signal associated with miRNA expression was seldom colocalized with a signal associated with mRNA expression (observed only in 3 of 49 loci). One robust colocalization was detected between a miRNA-eQTL for miR-4707–3p expression and GWAS signals for educational attainment and brain size phenotypes. Experimental manipulation of miR-4707–3p expression within primary human neural progenitors showed miR-4707–3p decreased the expression of predicted target genes and increased the number of proliferating cells. This example highlights the utility of miRNA-eQTLs in understanding how genetic variation may influence brain-related traits through regulation of miRNA expression.

## Results

### MicroRNA expression profiling

We profiled the expression of miRNAs across 223 fetal cortical tissue samples from donors between 14 and 21 gestation weeks using high-throughput small-RNA-sequencing. We used a specialized miRNA quantification algorithm, implemented in the miRge 2.0 package (*Lu et al., 2018*), to measure

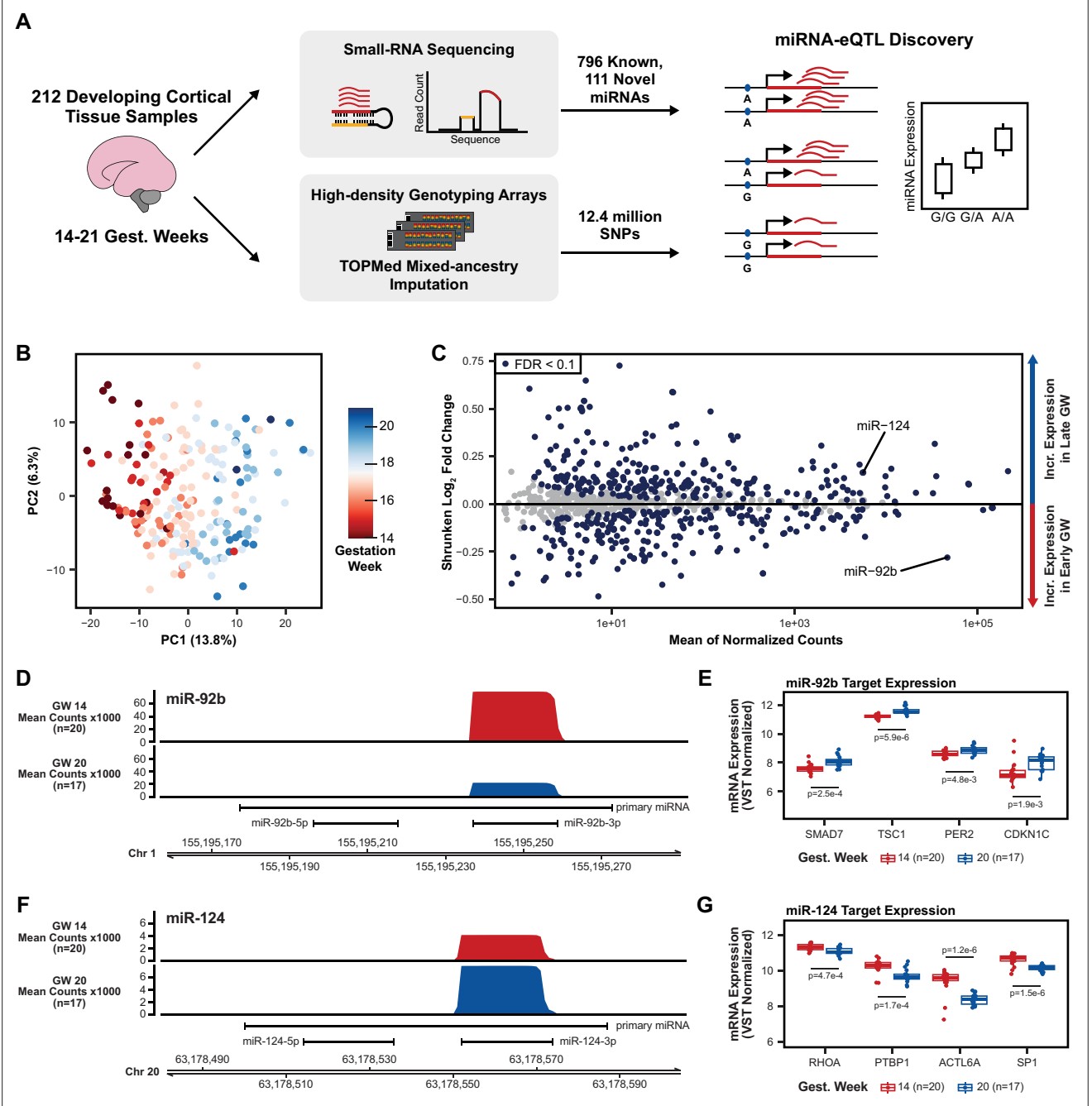

**Figure 1.** Study design and miRNA expression analysis. (**A**) Small-RNA sequencing was used on a total of 212 cortical tissue samples between 14 and 21 gestation weeks to identify novel miRNAs and quantify expression of known and novel miRNAs. Combined with imputed genotypes, a genetic association analysis was performed to discover local-miRNA-eQTLs. (**B**) Principal component analysis (PCA) on miRNA expression (miRBase release 22) after correcting for the known technical batch effects of sequencing pool and RNA purification method and removal of outlier samples (*Figure 1—figure supplement 1*). (**C**) Differential miRNA expression analysis using gestation week as the comparison variable. MiR-124, with known roles in neurogenesis, is upregulated among late gestation week samples, miR-92b, with known roles in progenitor proliferation, is upregulated among early gestation week samples. miRNAs with an FDR adjusted p-value <0.1 were considered significantly differentially expressed. (**D**) Coverage plot of mean small-RNA sequencing read counts at the miR-92b locus on chromosome 1 for a set of early gestation week samples (gest. week 14, n=20) and a set of late gestation week samples (gest. week 20, n=17). (**E**) Known miR-92b targets exhibit expression patterns consistent with down-regulation by a miRNA when separated between early and late gestation week samples. p-Values using a two-sided t-test between early and late gestation week samples. (**F**) Coverage plot, as in D, but for the miR-124 locus on chromosome 20. (**G**) Known miR-124 targets expression patterns as in E. p-values comparing mean expression as in E.

The online version of this article includes the following figure supplement(s) for figure 1:

*Figure 1 continued on next page*

*Figure 1 continued*

**Figure supplement 1.** Tissue sample miRNA expression and sample ancestry.

**Figure supplement 2.** Evidence for novel miRNAs.

the expression of known miRNAs cataloged in miRBase release 22 (March 2018; *Kozomara et al., 2019*). Combined with total-RNA-sequencing using an rRNA depletion-based library preparation in these same samples, collected in a previous study (*Walker et al., 2019*), we used rigorous quality control criteria (Methods) to eliminate 11 samples that were possible swaps, mixtures, or expression outliers (*Figure 1—figure supplement 1A–D*). Following batch effect correction for known technical confounding variables, a principal component analysis (PCA) on miRNA expression across the 212 remaining samples revealed the primary variation between samples was driven by gestation week at time of sample collection (*Figure 1B*). This finding is consistent with tissue composition differences between tissue from early gestation, composed of a greater proportion of neural progenitor cells, and late gestation, composed of a greater proportion of neurons (*Fernández et al., 2016*; *Nowakowski et al., 2016*). This analysis shows that after controlling for known technical variation, miRNA expression captures expected biological variability across these samples.

We conducted a differential expression analysis to identify the miRNAs changing across gestation weeks (*Figure 1C*). We found 269 miRNAs positively correlated and 246 miRNAs negatively correlated with gestation week (false discovery rate (FDR)<10%, *Supplementary file 2*). Examples include miR-92b, which has known roles in maintaining stem cell proliferation and is higher expressed in neural progenitors relative to differentiated neurons (*Roese-Koerner et al., 2017*; *Nowakowski et al., 2013*). By contrast, miR-124 is known to be higher expressed in post-mitotic neurons relative to neural progenitors and plays a role in promoting neuronal differentiation (*Sun et al., 2015*; *Makeyev et al., 2007*). Consistent with these roles, we observed miR-92b was significantly upregulated in early gestation week samples, and miR-124 was significantly upregulated in late gestation week samples (*Figure 1D and F*). Furthermore, validated mRNA targets of miR-92b (SMAD7, TSC1, PER2, and CDKN1C) show differential expression between early and late gestation week samples consistent with targeting and downregulation of mRNA by a miRNA (*Figure 1E*; *Sengupta et al., 2009*; *Wu et al., 2013*; *Long et al., 2017*; *Zhuang et al., 2016*; *Zhou et al., 2018*; *Lee et al., 2019*). Validated mRNA targets of miR-124 (RHOA, PTBP1, ACTL6A, and SP1) also show consistent expression patterns in early and late gestation week samples (*Figure 1G*; *Sun et al., 2015*; *Wang et al., 2016*; *Xue et al., 2016*; *Mondanizadeh et al., 2015*; *Dash et al., 2020*).This differential expression analysis and the expression patterns of known miRNA and mRNAs are expected given known cell-type compositions of cortical tissue during neuronal differentiation over developmental time.

In addition to quantifying the expression of known miRNAs in miRBase release 22, we quantified the expression of recently discovered miRNAs from studies by *Friedländer et al., 2014* (72 miRNAs from the Friedländer dataset were detected in this study) and (*Nowakowski et al., 2018*) (7 miRNAs from the Nowakowski dataset were detected in this study). Finally, using two annotation packages, miRge 2.0 (*Lu et al., 2018*) and miRDeep2 (*Friedländer et al., 2012*), we discovered 111 putatively novel miRNAs that were not previously annotated in miRBase release 22, Friedländer et al, or Nowakowski et al (*Supplementary file 1*). Novel miRNAs discovered in this study showed sequencing read coverage plots consistent with known miRNAs (*Figure 1—figure supplement 2A*). Furthermore, many novel miRNAs were differentially expressed between early and late gestation week samples (*Figure 1—figure supplement 2B*). This represents a novel resource of miRNAs that may not have been previously detected due to unique expression of these miRNAs in developing brain tissue or lower read depth and sample size obtained in previous studies. Although these novel miRNAs have characteristic sequencing read patterns consistent with known miRNAs, they require validation to demonstrate they function as miRNAs (*Moore et al., 2014*).

## Local-miRNA-eQTLs

Genotyping information from each of the 212 remaining donors revealed a sample of diverse ancestry (*Figure 1—figure supplement 1E*). Following TOPMed mixed-ancestry imputation, 12.4 million genetic variants were associated with the expression of 907 known and novel miRNAs across 212 fetal cortical tissue samples to perform a local-miRNA-eQTL analysis (*Figure 1A*; *Das et al., 2016*; *Taliun et al., 2021*). To control for population stratification in our association testing, we used a mixed-effects

linear model which included a kinship matrix as a random effect and 10 genotype principal components (PCs) as fixed effects (*Kang et al., 2010*; *Yang et al., 2014*). We included 10 miRNA-expression PCs, the technical variables of sequencing pool and RNA integrity number (RIN score), and the biological variables of sex and gestation week as additional fixed effect covariates in the model.

Following stringent local and global multiple testing correction (Methods), we identified 70 miRNAs with a local-eQTL, hereafter referred to as emiRs, using a hierarchical multiple comparisons

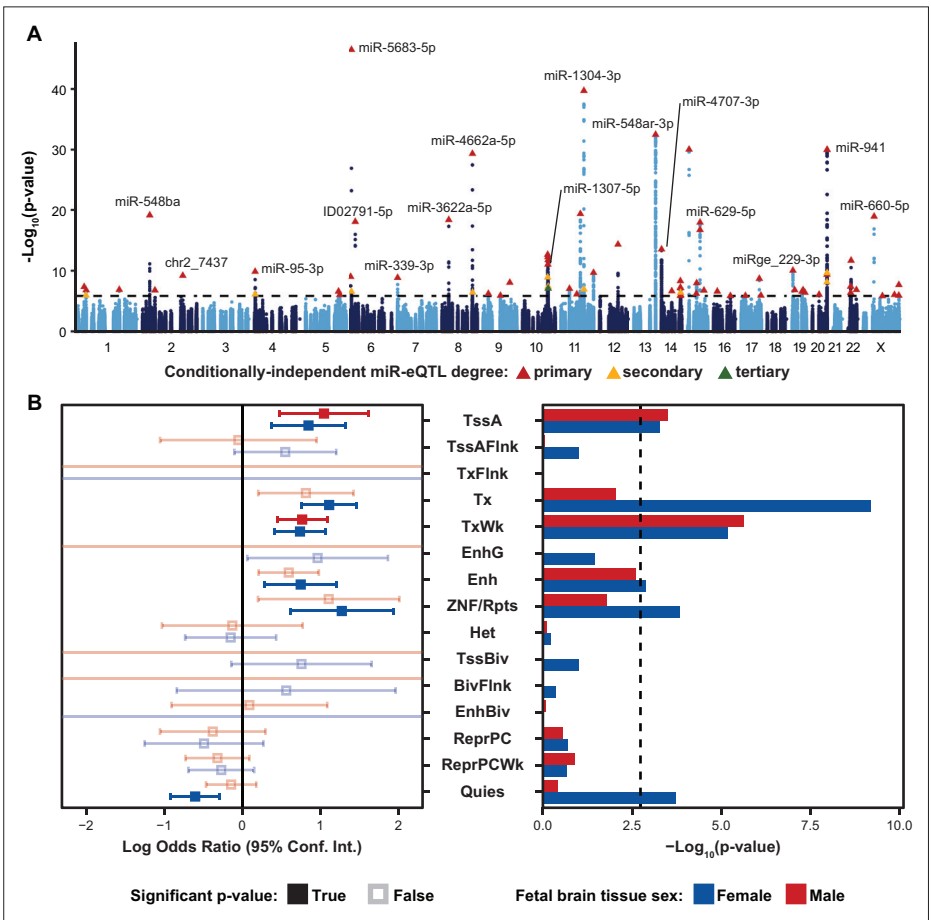

**Figure 2.** Local-miRNA-eQTLs and their enrichment in functional chromatin. (**A**) Manhattan plot showing -Log$_{10}$(p-values) for variants in a 2 Mb window around each expressed miRNA. Index variants for 70 primary, 14 secondary, and 1 tertiary eQTL denoted by triangles and colored by degree by which the eQTL is conditionally independent. Variants tested for association to more than one expressed miRNA are represented by independent points for each association p-value. Dotted line represents a stringent p-value significance threshold of $1.434 \times 10^{-6}$ after local and global multiple testing correction at FDR <5%. (**B**) Enrichment of local-miRNA-eQTLs within functionally annotated chromatin states. Local-miRNA-eQTLs passing the stringent p-value significance threshold were used for enrichment analysis. For other p-value significance thresholds, see *Figure 2—figure supplement 1*. Chromatin states were predicted using ChromHMM for both male and female fetal brain tissue. Left, log odds ratio and 95% confidence interval for significant enrichments are in solid colors. Non-significant enrichments in faded color outlines. Right, -Log$_{10}$(p-values) for each enrichment test. Dotted line represents a multiple testing corrected p-value threshold of $1.83 \times 10^{-3}$ (Bonferroni correction at α<0.05). miRNA-eQTLs are significantly enriched within active transcription start sites (TssA) and chromatin associated with strong transcription (Tx), weak transcription (TxWk), enhancers (Enh), and ZNF genes and repeats (ZNF/Rpts). There is also a significant depletion of miRNA-eQTLs within quiescent chromatin (Quies). Other abbreviations: flanking active tss (TssAFlnk), transcription at gene 5' and 3' (TxFlnk), genic enhancers (EnhG), heterochromatin (Het), bivalent/poised TSS (TssBiv), flanking TssBiv (BivFlnk), bivalent enhancer (EnhBiv), repressed polycomb (ReprPC), and weak repressed polycomb (ReprPCWk).

The online version of this article includes the following figure supplement(s) for figure 2:

**Figure supplement 1.** Characteristics of miRNA-eQTLs.

**Figure supplement 2.** Enrichment in functional chromatin at various thresholds.

threshold (FDR <5%, see Methods). Of these primary eQTLs, we identified an additional 14 loci with secondary eQTLs and one tertiary eQTL for a total of 85 conditionally independent local-miRNA-eQTLs (*Figure 2A*). Of the 70 emiRs, one miRNA was cataloged in Friedländer et al, two in Nowakowski et al, eight were novel miRNAs discovered within this fetal tissue dataset, and the remaining were previously cataloged in miRBase release 22 (*Supplementary file 3*). Two local-miRNA-eQTL eSNPs were associated with expression of two different emiRs, while the remaining eQTLs were uniquely associated with a single mature miRNA's expression (*Figure 2—figure supplement 1A*). Furthermore, the percent of expression variance explained by each primary miRNA-eQTL ranged from 10.5 to 65.2% (mean = 21.2%) similar to mRNA-eQTLs in the same tissue (*Figure 2—figure supplement 1B*; *Aygün et al., 2022*). To assess enrichments in functionally annotated genomic regions, we additionally used a relaxed, global-only, multiple testing correction threshold (see Methods) which increased the number of local-miRNA-eQTLs to 200 across 153 emiRs (153, 30, 13, 3, and 1 eQTLs of degree primary, secondary, tertiary, quaternary, and quinary, respectively). Discovery of these local-miRNA-eQTLs shows that genetic variation influences miRNA expression in the developing cortex, including the expression of previously unannotated miRNAs.

To characterize whether these miRNA-eQTLs are found in functionally annotated regions of the genome, we assessed whether eQTL signals were enriched in chromatin annotations that were from fetal tissue that were previously separated into male and female sexes (*Kundaje et al., 2015*). We identified significant enrichments of miRNA-eQTLs within active transcription start sites (TssA) and chromatin associated with strong transcription (Tx), weak transcription (TxWk), enhancers (Enh), and ZNF genes and repeats (ZNF/Rpts). There was also a significant depletion of miRNA-eQTLs within

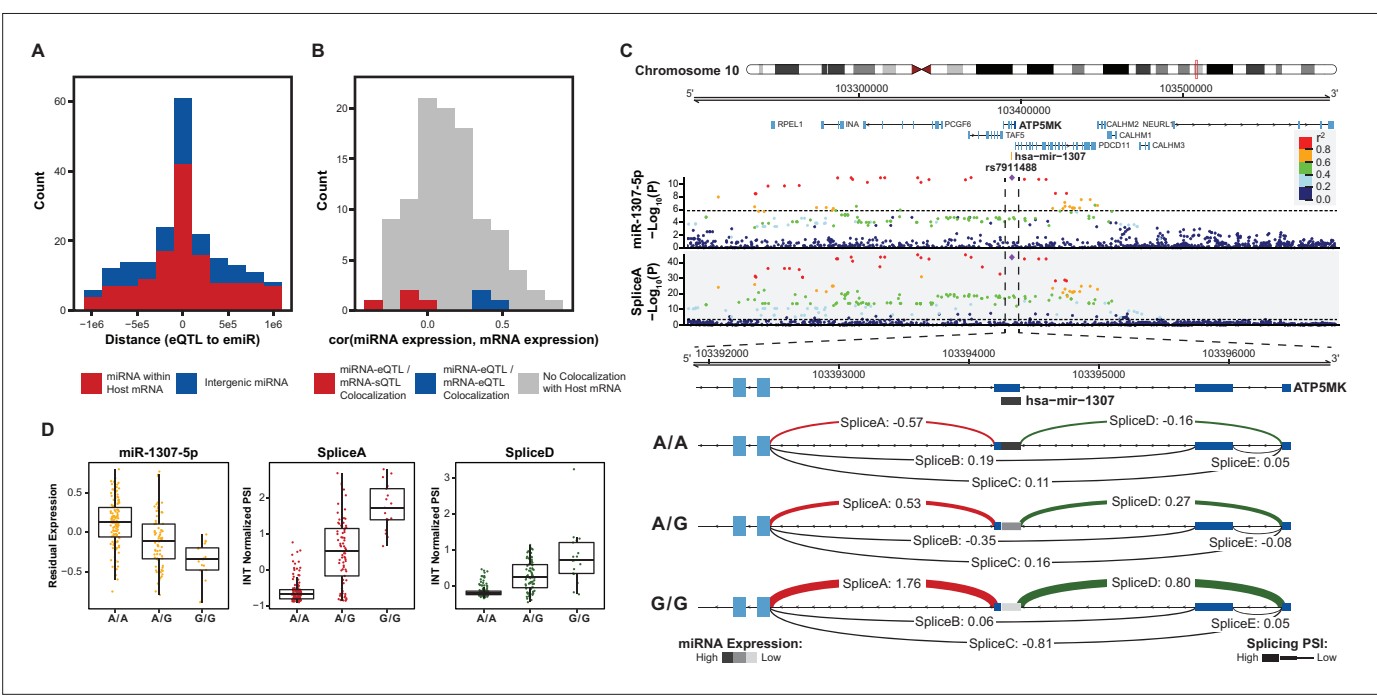

**Figure 3.** Colocalization of miRNA-eQTLs and mRNA-e/sQTLs. (**A**) Histogram of distances from miRNA-eQTL to its significantly associated miRNA (eSNP to emiR distance in base pairs) for each of the 85 miRNA-eQTLs across 70 emiRs. Colored by whether the emiR resides within a host mRNA or intergenic. (**B**) For emiRs that reside within a host mRNA, histogram of expression correlation between miRNA and mRNA host. Colors represent correlations for miRNA-eQTLs that colocalize with mRNA-eQTLs or mRNA-sQTLs. (**C**) Example of a miRNA-eQTL, for miR-1307–5p expression, which colocalizes with a mRNA-sQTL for splice variants in the 5' UTR of ATP5MK. Top locus dot-plot indicates -Log$_{10}$(p-values) on variant associations to miR-1307–5p expression. Bottom locus dot-plot shows -Log$_{10}$(p-values) for SpliceA utilization measured as inverse normal transformed (INT) percent spliced in (PSI). Variants are colored by pairwise linkage disequilibrium (LD, r$^2$) to the index variant, rs7911488. Zoom-in shows the 5' UTR of ATP5MK and the INT normalized PSI for each of five splice sites, labeled A-E, separated by genotype at rs7911488. Lines representing splice junctions at SpliceA and SpliceD are colored and weighted for relative PSI at the indicated genotypes. The line representing mir-1307 is shaded based on relative miR-1307–5p expression at the indicated genotypes. Dotted line in top and bottom locus dot-plot represents a stringent p-value significance threshold of 1.434x10$^{-6}$ and 2.957x10$^{-4}$, respectively, after local and global multiple testing correction at FDR <5%. (**D**) Residualized expression (after removal of known and unknown confounders) of miR-1307–5p at rs7911488 genotypes. INT normalized PSI values for SpliceA and SpliceD at rs7911488 genotypes.

quiescent chromatin (Quies; *Figure 2B*). These enrichments were robust to either stringent or relaxed multiple testing correction methods used to declare significant eQTLs (*Figure 2—figure supplement 2*). Enrichment of miRNA-eQTL signals within transcribed chromatin is expected given that most miRNAs are found within hosts or immediately adjacent to genes (*Liu et al., 2018*).

## Colocalization of miRNA-eQTLs with mRNA-e/sQTLs

Since over 50% of miRNAs are found within host genes, we classified the miRNA-eQTLs based on whether the emiR is located within a host gene or intergenic (*Liu et al., 2018*). Of the 70 emiRs at the stringent significance threshold, 49 are located within a host gene (100 of 153 emiRs are within hosts using the relaxed threshold). We found that miRNA-eQTLs are often close to their emiRs, and this trend is consistent whether or not the emiR is within a host gene (*Figure 3A*).

To further characterize these miRNA-eQTLs, we conducted a colocalization analysis to discover if the same genetic variants regulating miRNA expression also regulate mRNA expression and splicing. mRNA-eQTLs and mRNA-sQTLs were discovered in an expanded set of fetal cortical tissue samples largely overlapping with those samples used in our miRNA-eQTL analysis (*Aygün et al., 2021*; *Walker et al., 2019*). We found 17 and 12 colocalizations with eQTL and sQTLs, respectively (*Supplementary file 4*). For emiRs within a host mRNA without evidence of colocalization, we observed that miRNA expression was often positively correlated with mRNA expression (*Figure 3B*). Of these emiRs within hosts, we found 3 with mRNA-eQTL colocalizations and 4 with mRNA-sQTL colocalizations. Interestingly, expression of the few emiRs with a co-localized mRNA-eQTL were positively correlated with expression of their mRNA hosts, while expression of the few emiRs with a co-localized mRNA-sQTL were negatively correlated with their host mRNA expression.

This phenomenon is highlighted by a colocalization between a miRNA-eQTL of miR-1307–5p with an mRNA-sQTL for ATP5MK (*Figure 3C*). Hsa-mir-1307 sits within exon three of the 5' UTR of ATP5MK. In our dataset, we found evidence for five distinct intron excisions within the 5' UTR

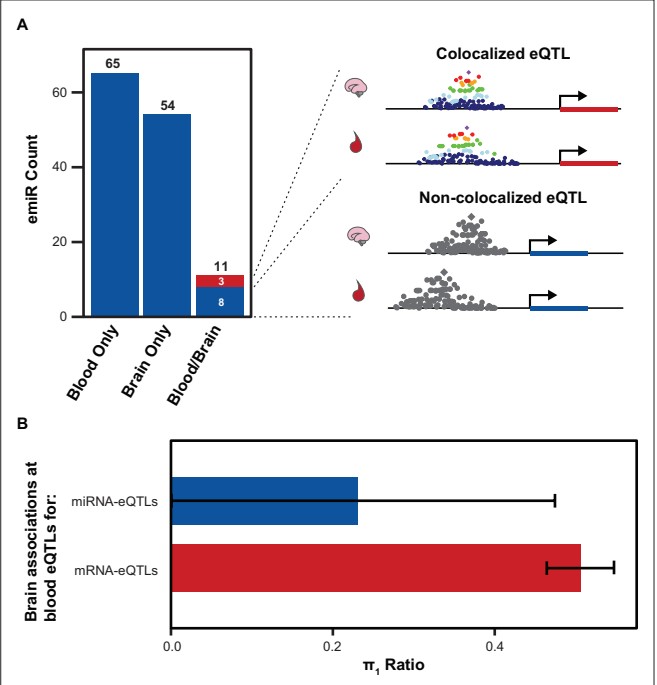

**Figure 4.** Comparison of Brain and Blood miRNA-eQTLs. (**A**) MiRNAs with significant associations, emiRs, separated by unique to blood, unique to brain, or shared. Three of 11 shared emiRs have co-localized genetic signals in the blood and brain datasets. Cartoon locus dot-plot shows a representation of a colocalized genetic signal or a non-colocalized signal for a miRNA-eQTL that is present in both tissues. (**B**) The fraction of brain eQTL associations that are estimated non-null associations within the blood eQTL dataset ($\pi_1$) separated by miRNA-eQTLs and mRNA-eQTLs. Error bars represent 95% upper and lower confidence intervals after 100 bootstrap samplings.

of ATP5MK (labeled SpliceA-SpliceE). Intron excision was quantified as percent spliced in (PSI) and normalized across all junctions in this cluster. Among these splice sites, we observed an association between genotypes (rs7911488 alleles A/G) and splice site utilization at SpliceA and SpliceD. This same variant was associated with expression of miR-1307–5p (*Figure 3D*). The G allele was associated with an increased PSI of SpliceA and SpliceD into the mRNA, while these same samples showed a decreased expression of miR-1307–5p. These data are consistent with biogenesis of miR-1307–5p from an exon of its host gene, ATP5MK. Increased miR-1307–5p expression resulted in fewer mRNA molecules which included exon three of ATP5MK. In this case, common genetic variation influences on both splicing and miRNA expression led to an understanding of the miRNA biogenesis.

## miRNA-eQTL tissue specificity

We next sought to quantify the degree to which our fetal cortical tissue miRNA-eQTLs are distinct from miRNA-eQTLs discovered in other tissues. We compared our developing brain miRNA-eQTLs (70 emiRs at the stringent testing correction threshold) to those of a large miRNA-eQTL analysis in blood (*Huan et al., 2015*). Of the 76 and 70 total emiRs in blood and brain tissue respectively, most are unique to a given tissue (65 and 54 are unique to blood and brain, respectively; *Figure 4A*). There are only 11 miRNAs that are emiRs in both blood and brain tissue. Of these emiRs present in both blood and brain, only three of these eQTLs have a colocalized genetic signal, implying the same causal variants affect miRNA expression in both tissues. The remaining eight emiRs found in both tissues do not share causal variants and therefore have tissue-specific genetic mechanisms regulating expression of these miRNAs. This shows that miRNA-eQTLs between blood and brain are highly tissue-specific, and the genetic variants regulating expression of shared emiRs can also be tissue-specific.

To further characterize the tissue-specificity of miRNA-eQTLs, we calculated the fraction of brain variant/miRNA pairs that are non-null associations within blood miRNA-eQTLs ($\pi_1$)(*Storey and*

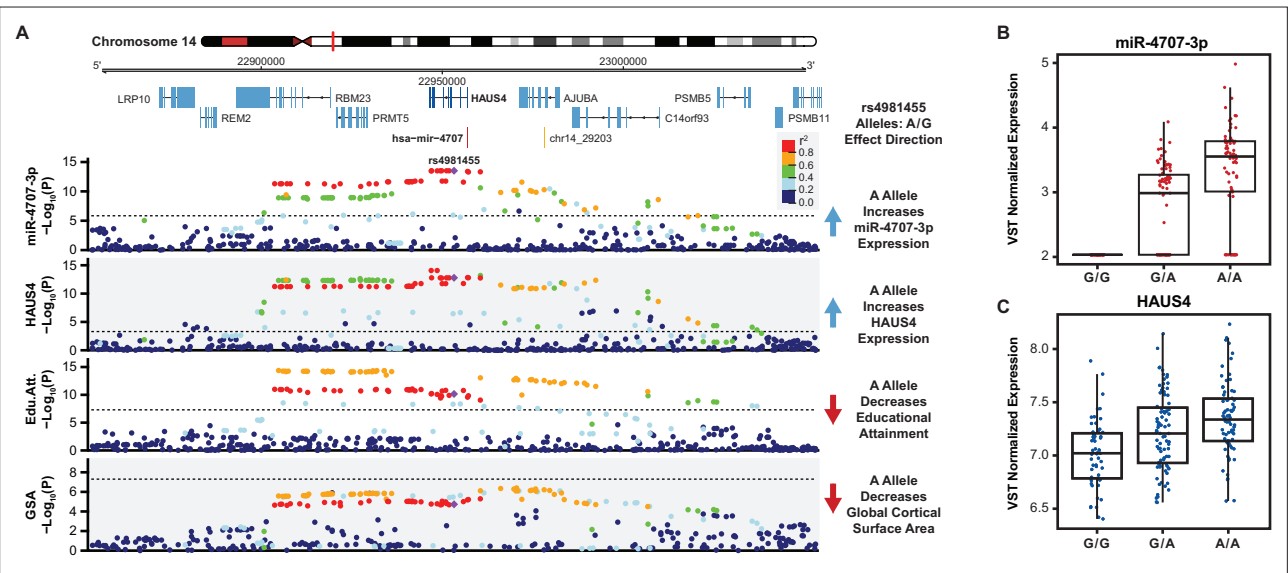

**Figure 5.** A miRNA-eQTL for miR-4707–3p colocalizes with multiple brain traits. (**A**) A locus dot-plot surrounding hsa-mir-4707 on chromosome 14 showing -Log₁₀(p-value) for various associations. Top track shows associations for expression of miR-4707–3p with the index variant, rs4981455. The second track shows associations surrounding a significant mRNA-eQTL to HAUS4 expression. The third track shows associations at this locus for a GWAS to educational attainment. The fourth track shows variant associations to global cortical surface area (GSA). Variants are colored by pairwise linkage disequilibrium (LD, $r^2$) to rs4981455 in each of the four tracks. Dotted line in the first and second tracks represent a stringent p-value significance threshold of $1.43 \times 10^{-6}$ and $8.17 \times 10^{-4}$ respectively after local and global multiple testing correction at FDR <5%. Dotted lines in bottom two tracks represents the global genome-wide significance threshold of $5 \times 10^{-8}$. (**B**) Boxplot showing variance stabilizing transformed (VST) expression of miR-4707–3p for samples with the indicated genotype at rs4981455. Samples with the G/G genotype show no expression of miR-4707–3p. (**C**) Boxplot showing VST expression of HAUS4, the host gene for mir-4707, at rs4981455.

The online version of this article includes the following figure supplement(s) for figure 5:

**Figure supplement 1.** MiR-4707–3p Alternate Allele Expression.

**Figure supplement 2.** Conditional analysis show miR-4707–3p colocalizes with educational attainment.

*Tibshirani, 2003*). Of the 76 blood emiRs, 52 were expressed at sufficient levels within our brain samples to test for genetic association. The fraction of blood associations found to be non-null in brain was 0.23 (+/-0.2 95% conf. int.)(*Figure 4B*). By comparison, the fraction of mRNA-eQTLs from blood tissue that are non-null associations in brain was 0.51 (+/-0.04 95% conf. int.)(*GTEx Consortium, Laboratory et al., 2017*). This provides further evidence that miRNA-eQTLs have stronger tissue specificity as compared to mRNA-eQTLs.

## Colocalization of miR-4707-3p miRNA-eQTL with brain size and cognitive ability GWAS

To determine if developing brain miRNA-eQTLs explain molecular mechanisms underlying risk for brain disorders or inter-individual differences in brain traits, we performed colocalization analyses between our 85 stringently defined local-miRNA-eQTLs and 21 GWAS summary statistics (*Supplementary file 4*). We discovered one robust colocalization between an eQTL for miR-4707–3p and multiple brain phenotypes, including educational attainment, brain size, and a subthreshold association for cortical surface area (*Figure 5A*). The eQTL for miR-4707–3p expression (rs4981455, alleles A/G) also co-localizes with an mRNA-eQTL for HAUS4 expression in developing cortical tissue. Hsa-mir-4707 is located within the 5' UTR of the HAUS4 gene. Despite this, and in contrast to the above example for mir-1307, the allele associated with increased expression of miR-4707–3p is also associated with increased expression of its host gene, HAUS4 (*Figure 5B and C*). The index variant, rs4981455, is in high linkage disequilibrium (LD; $r^2$ >0.99) with another variant (rs2273626, alleles C/A) which is within the 'seed' sequence of miR-4707–3p (*Figure 5—figure supplement 1A*). The A allele at rs2273626, corresponding to index variant allele G, would most likely change miR-4707–3p targeting. However, we did not detect any miR-4707–3p expression in samples with the G/G genotype at the index variant, therefore we did not study altered targeting of miR-4707–3p-G because it is not expressed (*Figure 5B*, *Figure 5—figure supplement 1B*). This finding is unlikely to be caused by reference mapping bias, because the miRNA quantification algorithm we used, miRge 2.0, accounts for common genetic variants within mature miRNA sequences (Methods; *Lu et al., 2018*). We also performed an allele specific expression analysis for donors that were heterozygous at rs2273626 (*Figure 5—figure supplement 1C*). rs2273636-A showed consistently lower expression, providing further support for the detected miRNA-eQTL (p=3.63 × 10$^{-14}$, using a paired, two-sided t-test). We detected only one read containing the A allele in three rs2273626 heterozygote donors. This indicates that miR-4707–3p is either expressed at levels too low for detection or not at all in chromosomes harboring the rs4981455 G allele.

In addition to the HAUS4 mRNA-eQTL colocalization, the miRNA-eQTL for miR-4707–3p expression is also co-localized with a locus associated with educational attainment (*Figure 5A*; *Lee et al., 2018*). Conditioning the miR-4707–3p associations with the educational attainment index SNP at this locus (rs1043209) shows a decrease in association significance, which is a hallmark of colocalized genetic signals (*Figure 5—figure supplement 2A*; *Nica et al., 2010*; *Dobbyn et al., 2018*). Additionally, the significance of the variants at this locus associated with miR-4707–3p expression are correlated to the significance for their association with educational attainment (Pearson correlation = 0.898, p=5.1 × 10$^{-107}$, *Figure 5—figure supplement 2B*). To further test this colocalization, we ran Summary-data-based Mendelian Randomization (SMR) at this locus which found a single causal variant to be associated with both miR-4707–3p expression and educational attainment (p=7.26 × 10$^{-7}$; *Zhu et al., 2016*). Finally, the heterogeneity in dependent instruments test (HEIDI), as implemented in the SMR package to test for two causal variants linked by LD, failed to reject the null hypothesis that there is a single causal variant affecting both gene expression and educational attainment when using the mixed-ancestry samples in this study as the reference population (*P*=0.159). The HEIDI test yielded similar results when estimating LD with 1000 Genomes European samples (p=0.120). All this evidence points to a robust colocalization between variants associated with both miR-4707–3p expression and educational attainment despite the different populations from which each study discovered the genetic associations.

At this locus, the alleles associated with increased expression of miR-4707–3p are also associated with decreased educational attainment. We also highlight here associations to global cortical surface area (GSA)(*Grasby et al., 2020*). Although this locus does not have genome-wide significant associations to GSA, a pattern of decreased p-values within the same LD block associated with

miR-4707–3p expression implies that this locus may hold a significant association in future GSA GWAS with increased sample size. Variants associated with increased expression of miR-4707–3p are associated with decreased GSA. Supporting the association to cortical surface area, variants associated with increased miR-4707–3p expression are also co-localized with variants associated with decreased head size and intracranial volume, a phenotype highly correlated with cortical surface area (data not shown from a publication in-preparation as well as a recently published article; *Knol et al., 2020*; *Nawaz et al., 2022*). This evidence suggests that the genetic risk for decreased brain size and decreased cognitive abilities may be mediated through increased expression of miR-4707–3p in developing human cortex. Few publications imply a known function for miR-4707–3p, however, HAUS4 is known to play a role in mitotic spindle assembly during cell division and a potent regulator of proliferation (*Ghanbari et al., 2017*; *Lawo et al., 2009*; *Uehara et al., 2009*). Unifying these observations lead us to a hypothesis, consistent with the radial unit hypothesis (*Rakic, 1988*), whereby increased expression of miR-4707–3p may influence neural progenitor fate decisions during fetal cortical development ultimately leading to a decreased cortical surface area.

## miR-4707-3p modulates target gene expression and proliferation in phNPCs

Given the genetic evidence implicating miR-4707–3p in cortical development and size, we next asked whether experimentally increasing the expression of miR-4707–3p in primary human neural progenitor cells (phNPCs), which model human neurogenesis, influences gene expression, proliferation, or cell fate decisions (*Figure 6—figure supplement 1A*; *Stein et al., 2014*). Using lentiviral transduction, we exogenously expressed either mir-4707 and GFP (pTRIPZ-mir-4707) or only GFP (pTRIPZ-control) in phNPCs. The expression of the miRNA and fluorophore were under the control of a tetracycline response element inducible with doxycycline (DOX). We transduced phNPCs from two genetically distinct donors: Donor 54 (D54, genotype G/G at rs4981455) and Donor 88 (D88, genotype A/A at rs4981455) that were cultured in media with growth factors that retain the phNPCs in a proliferative state as well as DOX (*Liang et al., 2021*). PhNPCs derived from D88 (homozygous for the miR-4707–3p increasing allele) endogenously expressed miR-4707–3p at a greater level than phNPCs derived from D54 (homozygous for the miR-4707–3p decreasing allele), as expected given their genotypes (*Figure 6—figure supplement 1C*). After confirming overexpression of miR-4707–3p induced by the expression construct (*Figure 6—figure supplement 1B and C*), we measured proliferation using a 2-hr EdU pulse to label cells in S-phase of the cell cycle. At 8 days post-transduction, we observed an increase in the number of EdU-positive nuclei in samples which over-expressed miR-4707–3p, which indicates this miRNA causes increased proliferation (*Figure 6—figure supplement 1C and D*). Increases in proliferation could be due to either more neurogenic or more self-renewal fate decisions. Investigating further, we measured gene expression via qPCR for a set of proliferation markers, progenitor markers, and neuronal markers in a time course experiment in phNPCs transduced with our expression construct (*Figure 6—figure supplement 1E*). At 4, 6, and 8 days post transduction, we observed increased expression of the proliferation markers, MKI67 and CCND1, which corroborate our findings from the EdU assay. We also observed an increase in the progenitor markers, PAX6 and SOX2, as well as the neuronal markers, DCX and TUJ1 (Beta-Tubulin III). Since the media used in these phNPC experiments contains growth factors which prevent differentiation, this experiment may have limited interpretability regarding how miR-4707–3p impacts neurogenesis.

We next sought to better characterize miR-4707–3p's effect on neurogenesis by modulating its expression in phNPCs grown in media where proliferative growth factors were removed, allowing differentiation (*Liang et al., 2021*; *Stein et al., 2014*). Using D54 phNPCs, we created DOX-inducible stable lines expressing either mir-4707 and GFP or only GFP using the same lentiviruses described above. We assayed the differentiating cells at 1 and 8 weeks post plating in differentiation media with DOX (*Figure 6A*). At both 1 and 2 weeks, miR-4707–3p was significantly upregulated in mir-4707 samples over control samples, demonstrating successful transduction (*Figure 6B*). To study the impacts of mir-4707 on mRNA expression in differentiating phNPCs, we assayed gene expression using microarrays at both week 1 and week 2. Following batch effect correction and removal of two outliers, PCA on the remaining 22 samples confirmed that they segregated primarily by expression vector (mir-4707 vs control) and time point (*Figure 6—figure supplement 2A–C*). Differential expression analysis between week 1 and week 2 control samples confirmed the in vitro culture system

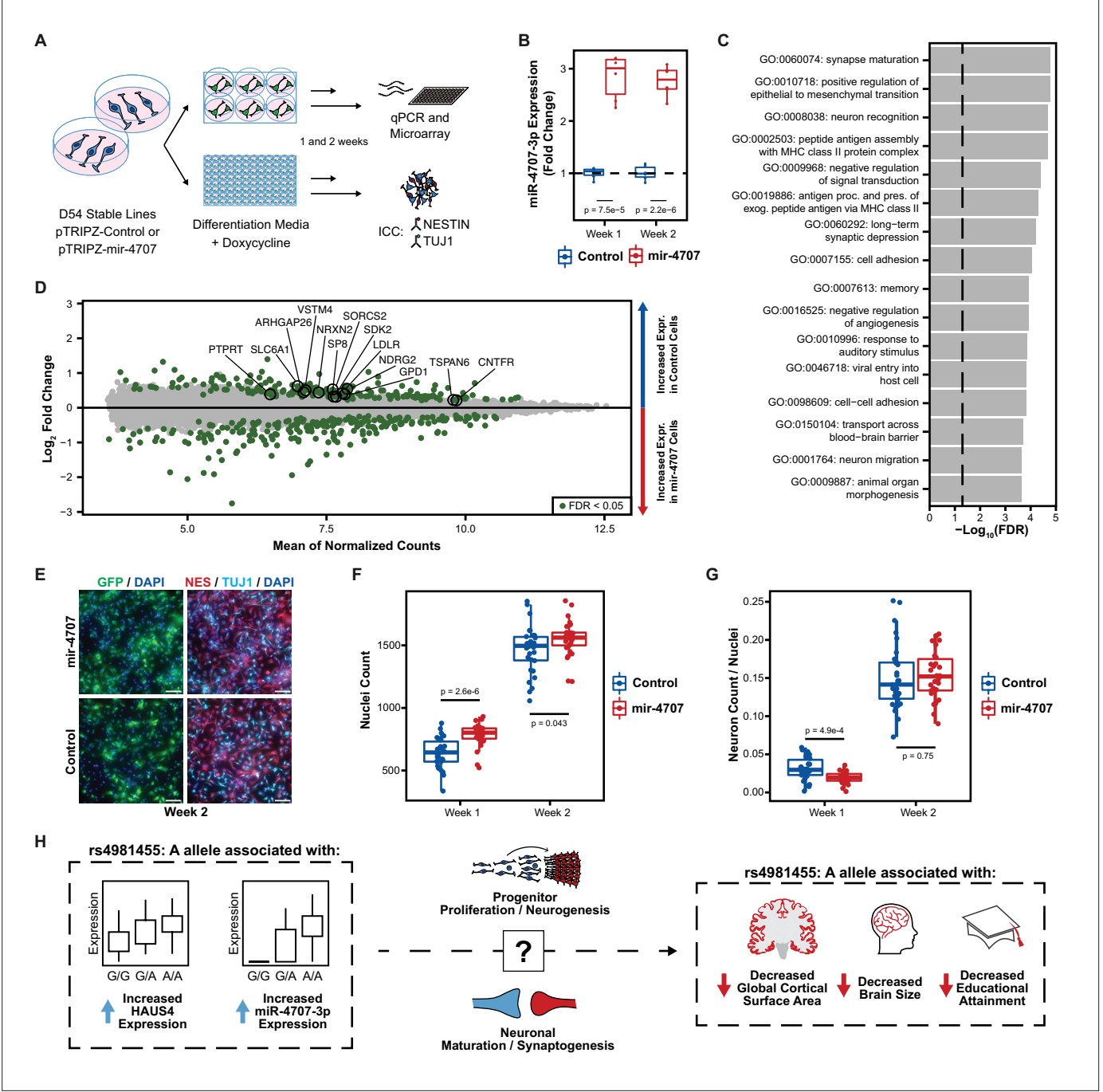

**Figure 6.** MiR-4707–3p overexpression in differentiating phNPCs. (**A**) Experiment overview. Primary human neural progenitor cells (phNPCs) from donor 54 (D54, genotype G/G at rs4981455) were transduced with pTRIPZ-mir-4707-EGFP (mir-4707) or a control pTRIPZ-EGFP and selected with puromycin to form stable lines in proliferation media. Stable lines were then plated in 6-well and 96-well plates into differentiation media with doxycycline to induce expression of miR-4707-3p+EGFP or control (EGFP only). At 1 and 2 weeks post plating in differentiation media, RNA was extracted from 6-well plates for qPCR and microarray experiments, and 96-well plates were fixed for ICC. (**B**) At 1 and 2 weeks post plating in differentiation media, D54 cells transduced with mir-4707 showed increased expression of miR-4707–3p as compared to control cell lines. p-Values from a two-sided t-test on 6 samples (wells) per condition. (**C**) Gene ontology analysis using all differentially expressed genes between mir-4707 and control cells at week 2 as seen in D. Enrichment p-values corrected by FDR. Dotted line represents an FDR corrected p-value of 0.05. (**D**) Differential mRNA expression between mir-4707 cells and control cells as measured by microarray assay at week 2. Positive log2 fold change indicates increased expression in control cells and decreased expression in mir-4707 cells. Selected genes are computationally predicted to be targets of miR-4707–3p and were down-regulated in mir-4707 cells at both 1 and 2 weeks. (**E**) Representative images from D54, week 2, mir-4707 and control cell lines stained with GFP, the neural progenitor marker NESTIN (NES), the neuronal marker TUJ1, and nuclear stain DAPI. Scale bars 100 μm. (**F**) DAPI stained nuclei per well at 1 and 2 weeks showed

*Figure 6 continued on next page*

*Figure 6 continued*

increased number of nuclei in mir-4707 cell lines as compared to control lines. p-Values from a two-sided t-test on 30 technical replicates (wells) per condition. (**G**) The fraction of neurons (cells labeled with higher TUJ1 expression as compared to NESTIN expression while also being GFP positive, see Methods) showed fewer neurons in mir-4707 lines at week 1 but no differences at week 2. p-Values from a two-sided t-test on 30 technical replicates (wells) per condition. (**H**) Summary of the genetic associations between rs4981455 and expression of HAUS4, expression of miR-4707–3p, global cortical surface area, brain size, and educational attainment. The molecular mechanism by which miR-4707–3p influences downstream traits may be mediated through neural progenitor fate decisions, neuronal maturation, or synaptogenesis. However, the colocalization of genetic associations between the molecular traits and the brain phenotypes may also be a result of confounding, pleiotropy, or other mechanisms yet to be investigated.

The online version of this article includes the following figure supplement(s) for figure 6:

**Figure supplement 1.** MiR-4707–3p overexpression in proliferating phNPCs.

**Figure supplement 2.** Microarray expression in differentiating phNPCs.

induces gene expression programs known to occur during differentiation (cell cycle down regulation and neurogenesis upregulation), as expected (*Figure 6—figure supplement 2D–F*).

We identified 1664 genes at week 1 and 430 at week 2 differentially expressed due to mir-4707 over-expression (*Supplementary file 6*). Of the miR-4707–3p targets predicted via TargetScan (*Agarwal et al., 2015*), 13 were consistently downregulated in mir-4707 samples as compared to control samples in both weeks 1 and 2 (*Figure 6D*). Among these are the gamma-aminobutyric acid (GABA) transporter, SLC6A1 (also known as GAT1), found in presynaptic terminals, NDRG2 and SORCS2, both with known roles in neurite outgrowth and branching, and TSPAN6, which interacts with glutamate receptors, mediates synaptic transmission and is associated with epilepsy and intellectual disability (*Goodspeed et al., 2020*; *Yin et al., 2020*; *Glerup et al., 2014*; *Salas et al., 2017*; *Becic et al., 2021*). These results suggest that miR-4707–3p has a direct effect on synapse maturation in differentiating human neural progenitors. Differentially expressed genes (both up- and down-regulated) at week 2 were enriched in ontologies associated with neuronal maturation including synapse maturation (GO:0060074), neuron recognition (GO:0008038), and negative regulation of signal transduction (GO:0009968; *Figure 6C*), providing further evidence that miR-4707–3p influences neuronal maturation phenotypes.

Immunocytochemistry (ICC) was then used to test the fraction of cells labeled with the progenitor marker, NESTIN, or the neuronal marker, TUJ1 (*Figure 6E*). MiR-4707–3p overexpression, at both 1 and 2 weeks, caused a significant increase in the number of nuclei per well as compared to control cells (*Figure 6F*), consistent with the previous results of increased proliferation due to miR-4707–3p overexpression (*Figure 6—figure supplement 1E*). When comparing the fraction of cells labeled as neurons (cells labeled with higher TUJ1 expression as compared to NESTIN expression, see Methods), we detected a lower percentage of neurons in week 1 mir-4707 samples and no differences in the percentage of neurons at week 2 (*Figure 6G*). These results show that miR-4707–3p decreases expression of predicted targets as well as genes relevant for synaptogenesis and increases neural progenitor proliferation. Although the decrease in neurogenesis at week 1 caused by miR-4707–3p is not consistent with the mechanism leading to cortical surface area decrease proposed by the radial unit hypothesis, our experiments nevertheless confirm that miR-4707–3p increases phNPC proliferation and modulates synaptic and neuronal maturation genes.

## Discussion

Using small-RNA-sequencing, we reveal robust miRNA expression across cortical tissue during mid-gestation, a stage and tissue which has not previously been captured in previous eQTL studies using standard RNA-sequencing techniques. In addition to the known roles of miR-92b and miR-124 on progenitor proliferation and neurogenesis, our differential expression analysis shows many other miRNAs likely play crucial roles in cortical development (*Nowakowski et al., 2013*; *Sun et al., 2015*). We were also able to find greater than 100 likely novel miRNAs as well as further evidence of recently discovered miRNAs within these tissue samples. These novel miRNAs have sequencing read coverage characteristic of known miRNAs, and they are differentially expressed across gestation weeks like miRNAs with known roles in neurogenesis. Investigating how these known and novel miRNAs function during neuronal differentiation may yield new gene regulatory mechanisms involved in human neurogenesis.

We also discovered 85 local-miRNA-eQTLs in a tissue-type and at a developmental time point with known influence on brain structure and cognitive traits. Despite many emiRs residing within host genes, miRNA-eQTLs seldom colocalize with mRNA-eQTLs for their host mRNAs. This implies a regulatory mechanism by which miRNA expression is largely independent of that which governs host mRNA expression, highlighting the unique information gained from miRNA-eQTLs that would otherwise be missed in standard mRNA-eQTL analyses. We found that the small subset of miRNA-eQTLs which colocalize with their host mRNA eQTLs have positively correlated expression, which would indicate a common genetic regulatory mechanism governing expression of both RNAs. On the other hand, miRNA-eQTLs which colocalize with a host mRNA-sQTL have negatively correlated expression. The genetic regulatory mechanisms governing miRNA biogenesis and expression uncovered by our eQTL analysis provides mostly unique mechanisms when compared to current mRNA-eQTL datasets. These miRNA-eQTLs will be a continued resource in the pursuit of describing the genetic risk loci uncovered by current and future GWAS for brain traits and disorders.

In contrast to mRNA based eQTLs, miRNA-based eQTLs appear to map less frequently to known brain disease loci. The lack of colocalizations are highlighted when compared to mRNA-eQTLs in the same tissue where 844 colocalizations with brain-trait and disorders were discovered across 18,667 mRNA-eQTLs despite a similar sample size (n=235 for mRNA-eQTLs vs n=212 for miRNA-eQTLs) (*Aygün et al., 2021*; *Walker et al., 2019*). This suggests that genetically regulated miRNAs either may not be a major contributor to neuropsychiatric disorders or current GWAS are underpowered to detect loci mediated through miRNA expression. We suspect that this may reflect the effect of purifying selection on miRNAs. MiRNAs are known to have broad downstream regulatory effects across hundreds or thousands of targeted mRNAs, and therefore the genetic mechanisms regulating miRNA expression may be more tightly regulated than for mRNA expression, as has previously been shown for transcription factors (*Agarwal et al., 2015*; *Battle et al., 2014*). Rare variants, less subject to the influences of selective pressure, may be governing miRNA expression which this study did not have the power or methodology (i.e. whole genome or exome sequencing) to detect.

We did find one colocalization between a miRNA-eQTL for miR-4707–3p expression and GWAS signals for brain size phenotypes and educational attainment. This revealed a possible molecular mechanism by which genetic variation causing expression differences in this miRNA during fetal cortical development may influence adult brain size and cognition (*Figure 6H*). Experimental overexpression of miR-4707–3p in proliferating phNPCs showed an increase in both proliferative and neurogenic gene markers with an overall increase in proliferation rate. At 2 weeks in differentiating phNPCs, we observed an overall increase in the number of cells upon miR-4707–3p overexpression, but we did not detect a difference in the number of neurons at this time point. Based on the radial unit hypothesis (*Rakic, 1988*; *Rakic, 2000*), we expected to find an overall decrease in proliferation or increase in neurogenesis upon miR-4707–3p overexpression which would explain decreased cortical surface area. However, our in vitro observations with phNPCs do not point to a mechanism consistent with the radial unit hypothesis by which increased miR-4707–3p expression during cortical development leads to decreased brain size. This has also been seen in similar studies using stem cells to model brain size differences linked with genetic variation (*Urresti et al., 2021*). Nevertheless, the transcriptomic differences associated with overexpression of miR-4707–3p in differentiating phNPCs suggest this miRNA may influence synaptogenesis or neuronal maturation, but these phenotypes may be better interrogated at later differentiation time points or with assays to directly measure neuronal migration, maturation, or synaptic activity.

An interesting feature of this genomic locus is the presence of both a miRNA-eQTL, for miR-4707–3p, and a mRNA-eQTL, for HAUS4. Although yet to be experimentally tested, the known effects on cell proliferation by the HAUS4 gene implies that increased expression of HAUS4 in neural progenitors would most likely lead to increased proliferation (*Lawo et al., 2009*; *Uehara et al., 2009*). It is not yet known whether miR-4707–3p and HAUS4 have similar or opposing influences on fate decisions during neurogenesis. Furthermore, we did not detect expression of miR-4707–3p in samples with the genotype G/G at the eQTL index variant rs4981455. This implies that the presence of the A allele turns on miR-4707–3p expression and individuals with the G/G genotype have no miR-4707–3p expression in developing cortical tissue. This further highlights the utility of studying miRNA-eQTLs, as uncovering only the mRNA-eQTL at this locus would not reveal the full genetic mechanism leading to inter-individual differences in the brain size and cognitive phenotypes.

Here we have proposed a biological mechanism linking genetic variation to inter-individual differences in educational attainment. Given the important societal implications education plays on health, mortality, and social stratification, a proposed causal mechanism between genes and education warrants greater scrutiny (*Braveman and Gottlieb, 2014*; *Zajacova and Lawrence, 2018*). Any given locus associated with educational attainment may be driven by a direct effect on brain development, structure, and function, an indirect genetic effect such as parental nurturing behavior, or confounding caused by discriminatory practices or societal biases (*Kong et al., 2018*; *Naqvi et al., 2021*). Given that expression was measured in prenatal cortical tissue, where confounding societal biases are less likely to drive genetic associations and that experimental overexpression of miR-4707 affected molecular and cellular processes in human neural progenitors, the evidence at this locus is consistent with a direct effect of genetic variation on brain development, structure, and function rather than being driven by confounding or indirect effects. However, there are some important caveats to this statement. First, our study only provides evidence for the direct effect on the brain at this one educational attainment locus. Our study does not provide evidence for the direct brain effects of any other locus identified in the educational attainment GWAS. Second, common variation at this locus explains a mere 0.00802% of the variation in educational attainment in a population, so this locus is clearly not predictive or the sole determinant of this phenotype. Third, the GWAS for educational attainment and brain structure were conducted in populations of European ancestry, and allele frequency differences at these loci cannot be used to predict differences in educational attainment or brain size across populations. Finally, although both experimental and association evidence suggests a causal link between this locus and educational attainment mediated through brain development, we are unable to directly test the influence of miR-4707–3p expression during fetal cortical development on adult brain structure and function phenotypes. Therefore, we cannot rule out the possibility that the causal mechanism between rs4981455 and adult cognition may be a result of genetic pleiotropy rather than mediation at this locus. Despite these caveats, identifying the mechanisms leading from genetic variation to inter-individual differences in educational attainment will likely be useful for understanding the basis of psychiatric disorders because educational attainment is genetically correlated with many psychiatric disorders and brain-related traits (*Lee et al., 2018*; *Matoba et al., 2020*).

In this study, we highlight one example of how individual differences in miRNA expression during fetal cortical development may lead to differences in brain size and cognition through altered neurogenesis. More work needs to be done to uncover the specific molecular pathways affected by miR-4707–3p, but the effect on cellular behavior is clear. The lengthening of neurogenesis and associated expansion of the brain are hallmarks of the evolutionary differences between humans and other mammals (*Lui et al., 2011*; *Sousa et al., 2017*; *Stepien et al., 2020*; *Otani et al., 2016*). Comparative genomic studies have revealed that human-specific gene regulatory differences in developing neocortex are associated with neurogenesis, brain complexity, and disease, and that primate-specific miRNAs have been shown to play a role in post transcriptionally regulating gene expression associated with these developmental processes (*Florio et al., 2017*; *Florio et al., 2018*; *Namba and Huttner, 2017*; *Hu et al., 2011*; *Barbash et al., 2014*). Here, we provide further evidence that miRNAs expressed during neocortical development may ultimately influence human cognition. Continued work on miRNA expression patterns and miRNA regulatory networks during fetal brain development will be crucial to further understand the molecular pathways that lead to brain size or cognitive differences between individuals.

## Methods

**Key resources table**

| Reagent type (species) or resource | Designation | Source or reference | Identifiers | Additional information |
|---|---|---|---|---|
| Biological sample (Homo-sapiens) | Prenatal cortical tissue | UCLA Gene and Cell Therapy Core | | |
| Commercial assay or kit | miRNeasy-mini | QIAGEN | 217004 | |
| Commercial assay or kit | TruSeq Small RNA Library Prep Kits | Illumina | RS-200 | |

*Continued on next page*

*Continued*

| Reagent type (species) or resource | Designation | Source or reference | Identifiers | Additional information |
|---|---|---|---|---|
| Commercial assay or kit | TruSeq Stranded RNA Library Prep Kits | Illumina | 20020597 | |
| Software, algorithm | miRge 2.0 | DOI https://doi.org/10.1186/s12859-018-2287-y | | |
| Software, algorithm | miRDeep2 | DOI https://doi.org/10.1093/nar/gkr688 | | |
| Software, algorithm | Bowtie | DOI https://doi.org/10.1038/nmeth.1923 | v1.2.2 | |
| Software, algorithm | featureCounts | DOI https://doi.org/10.1093/bioinformatics/btt656 | | |
| Software, algorithm | STAR | DOI https://doi.org/10.1093/bioinformatics/bts635 | v2.5.4b | |
| Software, algorithm | PLINK | DOI https://doi.org/10.1186/s13742-015-0047-8 | v1.9 | |
| Software, algorithm | McCarthy Group's HRC-1000G imputation preparation tool | https://www.well.ox.ac.uk/~wrayner/tools/ | | |
| Software, algorithm | Michigan Imputation Server | DOI https://doi.org/10.1038/ng.3656 | | |
| Software, algorithm | VerifyBamID | DOI https://doi.org/10.1016/j.ajhg.2012.09.004 | v1.1.3 | |
| Software, algorithm | DESeq2 | DOI https://doi.org/doi:10.18129/B9.bioc.DESeq2 | | |
| Software, algorithm | limma | DOI https://doi.org/doi:10.18129/B9.bioc.limma | | |
| Software, algorithm | EMMAX | DOI https://doi.org/10.1038/ng.548 | | |
| Software, algorithm | eigenMT | DOI https://doi.org/10.1016/j.ajhg.2015.11.021 | | |
| Software, algorithm | Summary-data-based Mendelian Randomization | DOI https://doi.org/10.1038/ng.3538 | | |
| Software, algorithm | GARFIELD | DOI https://doi.org/10.1038/s41588-018-0322-6 | v2 | |
| Software, algorithm | qvalue | DOI https://doi.org/doi:10.18129/B9.bioc.qvalue | | |
| Software, algorithm | RMA | DOI https://doi.org/10.1093/bioinformatics/btq431 | | |
| Software, algorithm | topGO | DOI https://doi.org/doi:10.18129/B9.bioc.topGO | | |
| Recombinant DNA reagent | pTRIPZ (plasmid) | ThermoFisher | RHS4750 | |
| Recombinant DNA reagent | pTRIPZ-EGFP (plasmid) | This paper | | RFP of original pTRIPZ replaced with EGFP |
| Recombinant DNA reagent | pTRIPZ-mir-4707-EGFP (plasmid) | This paper | | mir-4707 primary sequence cloned into pTRIPZ-EGFP |
| Recombinant DNA reagent | PAX2 (plasmid) | Addgene | 12260 | |
| Recombinant DNA reagent | pMD2.G | Addgene | 12259 | |
| Commercial assay or kit | qPCR Lentivirus Titration Kit | Applied Biological Materials | LV900 | |
| Commercial assay or kit | FUGENE HD | Promega | E2311 | |

*Continued on next page*

*Continued*

| Reagent type (species) or resource | Designation | Source or reference | Identifiers | Additional information |
|---|---|---|---|---|
| Cell line (Homo-sapiens) | HEK293T | ATCC | CRL-11268 | |
| Cell line (Homo-sapiens) | Donor 54 | This paper | D54 | primary human neural progenitor cells (male) |
| Cell line (Homo-sapiens) | Donor 88 | This paper | D88 | primary human neural progenitor cells (male) |
| Chemical compound, drug | DMEM | Life Technologies | 11995081 | |
| Chemical compound, drug | FBS | Sigma-Aldrich | F2442 | |
| Chemical compound, drug | Antibiotic-Antimycotic | Life Technologies | 15240096 | |
| Chemical compound, drug | Growth Factor Reduced Matrigel | Corning | 354230 | |
| Chemical compound, drug | Neurobasal A | Life Technologies | 10888–022 | |
| Chemical compound, drug | Primocin | Invivogen | ant-pm-2 | |
| Chemical compound, drug | BIT 9500 | Stemcell Technologies | 9500 | |
| Chemical compound, drug | Glutamax | Life Technologies | PHG0313 | |
| Chemical compound, drug | Heparin | Sigma-Aldrich | H3393-10KU | |
| Chemical compound, drug | EGF | Life Technologies | PHG0313 | |
| Chemical compound, drug | FGF | Life Technologies | PHG0023 | |
| Chemical compound, drug | LIF | Life Technologies | PHC9481 | |
| Chemical compound, drug | PDGF | Life Technologies | PHG1034 | |
| Chemical compound, drug | Trypsin-EDTA | Life Technologies | 25300062 | |
| Chemical compound, drug | Doxycycline | Sigma-Aldrich | D9891 | |
| Chemical compound, drug | B27 | Life Technologies | 17504–044 | |
| Chemical compound, drug | NT-3 | Life Technologies | PHC7036 | |
| Chemical compound, drug | BDNF | Life Technologies | PHC7074 | |
| Commercial assay or kit | Dnase | QIAGEN | 79254 | |
| Commercial assay or kit | iScript cDNA Synthesis Kit | Bio-Rad | 1708891 | |
| Commercial assay or kit | TaqMan Advanced miRNA cDNA Synthesis Kit | ThermoFisher | A28007 | |
| Commercial assay or kit | SsoAdvanced Universal SYBR Green Supermix | Bio-Rad | 1725271 | |

*Continued on next page*

*Continued*

| Reagent type (species) or resource | Designation | Source or reference | Identifiers | Additional information |
|---|---|---|---|---|
| Commercial assay or kit | TaqMan Advanced miRNA Assays | ThermoFisher | A25576 | |
| Commercial assay or kit | TaqMan Fast Advanced Master Mix | ThermoFisher | 4444557 | |
| Commercial assay or kit | Clariom S Pico Assay | ThermoFisher | 902963 | |
| Commercial assay or kit | Click-iT EdU Imaging Kit (AF647) | Invitrogen | C10340 | |
| Chemical compound, drug | Paraformaldehyde in PBS | FisherScientific | 50-980-487 | |
| Chemical compound, drug | Triton X-100 | Sigma-Aldrich | T8787 | |
| Chemical compound, drug | Goat serum | MP Biomedicals | 219135680 | |
| Chemical compound, drug | Tween-20 | FisherScientific | BP337500 | |
| Antibody | anti-GFP (chicken polyclonal) | FisherScientific | AB16901 | (1:500) |
| Antibody | anti-chicken-AF488 (goat polyclonal) | ThermoFisher | A11011 | (1:1000) |
| Antibody | anti-NESTIN (rabbit polyclonal) | Millipore | ABD69 | (1:500) |
| Antibody | anti-TUJ1 (mouse monoclonal) | BioLegend | 801202 | (1:1000) |
| Antibody | anti-rabbit-AF568 (goat polyclonal) | ThermoFisher | A11036 | (1:1000) |
| Antibody | anti-mouse-AF750 (goat polyclonal) | ThermoFisher | A21037 | (1:1000) |
| Chemical compound, drug | DAPI | ThermoFisher | 62248 | |

## Tissue procurement

Human prenatal cortical tissues samples were obtained from the UCLA Gene and Cell Therapy Core following institutional review board regulations for 223 genetically distinct donors (96 females, 127 males, 14–21 gestation weeks) following voluntary termination of pregnancy. This study was declared exempt by the UNC Institutional Review Board (16–0054). Donors with known aneuploidies were excluded. This sample size is consistent with previous eQTL studies (*GTEx Consortium, Laboratory et al., 2017*). Tissue samples were flash frozen after collection and stored at –80°C. These tissue samples overlapped with those used in a previous mRNA-eQTL study. Cortical tissue samples from an additional 3 donors were microdissected into germinal zone and cortical plate sections as previously described (*Lu et al., 2018*) yielding 17 more tissue samples which were used for novel miRNA discovery but withheld from the miRNA-eQTL analysis. The investigators were not explicitly blinded to the donor during cell culture or library preparation. However, the investigators did not have knowledge of specific donor genotypes while libraries were prepared or sequenced.

## Library preparation and sequencing

Total RNA was extracted using miRNeasy-mini (QIAGEN 217004) kits or was extracted using trizol with glycogen followed by column purification. Library preparation for small-RNA was conducted using TruSeq Small RNA Library Prep Kits (Illumina RS-200). RNA libraries were randomized into eight pools and run across eight lanes of an Illumina HiSeq2500 sequencer at 50 base-pair, single-end reads to a mean sequencing depth of 11.7 million reads per sample. mRNA library preparation and sequencing were previously described (*Walker et al., 2019*). Briefly, library preparation for total RNA was conducted via TruSeq Stranded RNA Library Prep Kits (Illumina 20020597) with Ribozero Gold ribosomal RNA depletion. Libraries were sequenced with 50 base-pair, paired-end reads to a mean read depth of 60 million reads per sample.

## MicroRNA expression analysis

Small RNA-sequencing FASTQ files were used as input to the miRge 2.0 (*Lu et al., 2018*) annotation workflow to quantify expression of known miRNAs from miRBase release 22 (March 2018; *Kozomara et al., 2019*). Briefly, sequencing reads were first quality controlled, adaptors removed, and collapsed into unique reads. The reads were then annotated against libraries of mature miRNA, hairpin miRNA, tRNA, snoRNA, rRNA, and mRNA. The miRge 2.0 workflow protects against reference mapping bias by incorporating common genetic variants into the mature miRNA sequence library and allowing zero mismatched bases on the first pass of read annotation. A second pass of unannotated reads allows for mismatched bases to identify isomiRs. Unmapped reads were then used as input to a second annotation pipeline to quantify expression of recently discovered novel miRNAs from *Friedländer et al., 2014* and *Nowakowski et al., 2018*. Bowtie v1.2.2 (*Langmead et al., 2009*) was used to map reads to the UCSC hg38 reference genome using the following options: -v 2–5 1–3 2 `--norc` -a `--best` -S `--chunkmbs` 512. Mapped reads were then counted using featureCounts (*Liao et al., 2014*) against a custom GTF file including the Friedländer and Nowakowski novel miRNAs using the following options: -s 0 M -f -O.

To test for allele-specific expression of miR-4707–3p containing variants at rs2273626, we used Bowtie with a modified reference genome which only included miR-4707–3p mature sequence. Sequencing reads were allowed to map to either AGCCCGCCCCAGCCGAGGTTCT (reference allele C, complementary strand G) or AGCCCTCCCCAGCCGAGGTTCT (alternate allele A, complementary strand T) with no mismatches using the Bowtie options: -n 0 `--norc --all` -S. Mapped reads were then counted with featureCounts as above. Read counts, specific to each genotype, from samples heterozygous at rs2273626 were plotted in *Figure 5—figure supplement 1C*.

We also defined an additional set of novel miRNAs discovered within our 240 sample dataset using miRge 2.0 and miRDeep2 (*Friedländer et al., 2012*) prediction pipelines. Putatively novel miRNAs predicted using miRge 2.0 (all predictions) and miRDeep2 (predictions with a score greater than zero) were removed if they overlapped with each other, known miRNAs from miRBase release 22, or recently discovered miRNAs from the Friedländer and Nowakowski datasets. Sequencing read coverage plots for each novel miRNA annotation were also created to visually inspect each annotation. Putatively novel miRNAs without a characteristic 5' and 3' mapping pattern (*Figure 1—figure supplement 2*) were removed. Uniquely novel annotations, which passed visual inspection, were compiled and used to create a custom GTF file for use in the above annotation pipeline to quantify novel miRNA expression. Read counts of miRNAs from miRBase release 22, Friedländer et al, Nowakowski et al, and putatively novel annotations from this study were combined into one count matrix for use in downstream analyses.

## mRNA expression analysis

Total RNA-sequencing FASTQ files were first filtered and adapter trimmed using trim_galore and the following options: `--length` 20 `--stringency` 5. Filtered and trimmed reads were mapped to GRCh38 using STAR v2.5.4b (*Dobin et al., 2013*). Mapped reads were counted using featureCounts (*Liao et al., 2014*) against the Ensembl GRch38.p7 human gene annotations using the following options: -T 4 p -t exon. Count data for each sample was combined into a count matrix for downstream analyses.

## Genotyping

Genomic DNA was isolated using DNeasy Blood and Tissue Kit (QIAGEN 69504), and genotyping was performed on either HumanOmni2.5 or HumanOmni2.5Exome (Illumina) platforms in eight batches. SNP genotypes were exported and processed into PLINK format using PLINK v1.9 (*Chang et al., 2015*). Quality control and pre-processing of genotypes was also done using PLINK v1.9 as previously described (*Liang et al., 2021*). Briefly, SNPs were filtered based on Hardy-Weinberg equilibrium, minor allele frequency, individual missing genotype rate, and variant missing genotype rate (plink `--hwe` 1e-6 `--maf` 0.01 `--mind` 0.1 `--geno` 0.05) yielding a total of 1,760,704 genotyped SNPs.

## Imputation

Sample genotypes were prepared for imputation using the McCarthy Group's HRC-1000G imputation preparation tool (https://www.well.ox.ac.uk/~wrayner/tools/): perl HRC-1000G-check-bim.pl -b

AllSamplesQC.bim -f AllSamplesQC.frq -r 1000GP_Phase3_combined.legend -g -p ALL. This tool produces a script to separate genotype files by chromosome, filter variants to include only reference SNPs, and convert the Plink filesets into VCF files. Compressed VCF files were then uploaded to the Michigan Imputation Server for use in the Minimac4 imputation pipeline (**Das et al., 2016**; **Fuchsberger et al., 2015**). TOPMed Freeze5 was used as the reference panel for imputation (**Taliun et al., 2021**). Imputed genotypes were filtered for an imputation quality score ($R^2$) greater than 0.3, Hardy-Weinberg equilibrium p-value greater than 1e-6, and a minor allele frequency greater than 0.01.

## Sample quality control

Sample sex was called from the genotype data using PLINK v1.9 based on X chromosome heterozygosity. Sex was confirmed by checking expression of XIST within the mRNA-sequencing data. Of the 223 samples, two were declared female using the genotype data but male by XIST expression and were excluded from downstream analysis. We further sought to detect sample swaps or mixtures by evaluating the consistency of genotypes called via genotyping and those that can be detected by RNA-sequencing using VerifyBamID v1.1.3 (**Jun et al., 2012**). We detected four samples that were mixtures or possible sample swaps with their assigned genotype data (FREEMIX >0.04 or CHIPMIX >0.04). Finally, five samples were identified as outliers via PCA analysis after accounting for known technical confounders (**Figure 1—figure supplement 1C**). A total of 212 samples (93 females, 119 males, 14–21 gestation weeks) were used in the miRNA-eQTL analysis.

## PCA and differential miRNA expression analysis

Principal component analysis (PCA) was performed using the prcomp() function within the stats package of the R software language (**R Development Core Team, 2021**). Only miRBase release 22 expressed miRNAs with at least 10 counts across at least 10 samples were included when doing PCA. MiRNA expression was first normalized using the variance-stabilizing transformation (VST) function within DESeq2 (**Love et al., 2014**). To correct for known batch effects (sequencing-pool and RNA-purification method; **Figure 1—figure supplement 1**), we used the removeBatchEffect() function within the limma package on the VST transformed expression matrix (**Ritchie et al., 2015**). While removing batch effects, we preserved the effect of gestation week using the design option within removeBatchEffect(). After known batch effects were removed, PCA was repeated to confirm removal of technical variation across samples and to identify expression outliers (**Figure 1—figure supplement 1C**).

Differential expression analysis was conducted on the 212 samples which survived quality control filtering using DESeq2 (**Love et al., 2014**). Expression of all known and novel miRNAs, which survived the above expression threshold, were used in the analysis. Gestation week was used as the treatment variable while controlling for technical confounding variables: sequencing-pool, RNA-purification method, RNA-integrity number (RIN), and RNA concentration after extraction. Differentially expressed miRNAs with a Benjamini-Hochberg adjusted p-value <0.1 were deemed significant (false discovery rate (FDR)<10%). For visualization of differentially expressed miRNAs (**Figure 1C**), $\log_2$(fold change) values were shrunk using the apeglm method (**Zhu et al., 2019**).

## local-miRNA-eQTL mapping

We conducted local-eQTL mapping using tissue samples from 212 donors. A total of 866 miRNAs with an expression of at least 10 counts across at least 10 samples were included in this analysis. Since expressed miRNAs can originate from more than one genomic locus, associations were conducted at a total of 907 genomic loci. MiRNA counts were normalized using a variance-stabilizing transformation function within DESeq2 (**Love et al., 2014**). Normalized expression values were adjusted using a linear model accounting for population stratification as well as known and unknown confounders. Known confounders included: sequencing pool, RNA purification method, rna integrity score (RIN), sex, and donor gestation week. Unobserved confounding variables on miRNA expression were controlled using the first 10 principal components from a PCA. Population stratification was controlled using the first 10 principal components from a genotype PCA using PLINK v1.9.

Association testing between variants and residual miRNA expression (after adjusting for confounders) was done using a linear mixed-effects model as implemented in the EMMAX package (**Kang et al., 2010**). To further control for cryptic relatedness and population stratification, we

included an identity-by-state kinship matrix, also constructed using EMMAX (emmax-kin -v -h -s -d 10) by excluding variants located on the same chromosome as the variants tested in the association analysis (MLMe method)(*Yang et al., 2014*). In order to prevent a single outlier from driving the association results, variants were filtered before association testing to include only those which did not have exactly one homozygous minor sample and the number of heterozygous samples were greater than one. Variants within 1 Mb upstream of the mature miRNA start position or 1 Mb downstream of the mature miRNA end position were tested for association. Using EMMAX, imputed variant dosages were used for association testing (emmax -v -d 10 -Z -t [variant_doages] -k [kinship_mat] -o [output_file] -p [expression_file]).

For multiple-testing adjustment, we employed a two-stage analysis which accounts for linkage-disequilibrium between the variants tested (local adjustment) and the total number of miRNAs tested (global adjustment). First, p-values were adjusted locally using the eigenMT package (*Davis et al., 2016*). Locally adjusted p-values were then further adjusted using the Benjamini-Hochberg multiple testing correction for a 5% false discovery rate (FDR <0.05) across the 907 genomic loci tested. This yielded a nominal p-value of $1.434 \times 10^{-6}$ as the stringent threshold for which variants were declared significantly associated with miRNA expression. We also declared a relaxed threshold using only global adjustment with a 5% FDR across the 6.3 million independent association tests which yielded a nominal p-value threshold of $2.804 \times 10^{-5}$.

To declare conditionally independent local-miRNA-eQTLs, primary eQTLs were first defined as the most significant variant/miRNA pair for each expressed miRNA with at least one variant below the given nominal p-value threshold. An emiR was defined as a miRNA that has at least one variant associated with it. For each emiR, variant association testing was repeated for variants within the original 2 Mb window with the genotypes of the primary eQTL added to the association equation. The most significant variants below the original nominal p-value threshold ($1.434 \times 10^{-6}$ for stringent or $2.804 \times 10^{-5}$ for relaxed) within this secondary analysis (if any) were defined as secondary eQTLs. The process was repeated (with the inclusion of primary and secondary genotypes in the association equation to find tertiary eQTLs, primary, secondary, and tertiary to find quaternary eQTLs, etc.) until no variants remained below the nominal p-value threshold.

## Colocalization analysis

Colocalization of local-miRNA-eQTLs with brain-relevant trait GWAS summary statistics (*Supplementary file 4*), blood local-miRNA-eQTLs (*Huan et al., 2015*), and fetal brain local-mRNA-e/sQTLs from a largely overlapping sample (*Walker et al., 2019*) was done by first finding overlapping variants with LD $r^2 \geq 0.8$ of each local-miRNA-eQTL index variant and the set of variants within LD $r^2 \geq 0.8$ of each trait GWAS, disorder GWAS, eQTL, or sQTL index variant. LD for local-miRNA-eQTL, local-mRNA-eQTL, and local-mRNA-sQTL variants were calculated using the genotypes of the mixed-ancestry samples within each study. LD for GWAS and blood cis-miRNA-eQTL variants were calculated from 1000 Genomes phase 3 European genotypes (*1000 Genomes Project Consortium et al., 2015*). After overlaps were detected, colocalization was confirmed by conditional analysis which incorporated the genotypes for a given overlapping index variant into the local-miRNA-eQTL association equation. A resultant increase in p-value beyond the stringent or relaxed p-value threshold confirmed a colocalization between the local-miRNA-eQTL index variant and the given trait GWAS, disorder GWAS, or QTL index variant. At the miR-4707–3p colocalization to educational attainment, Summary-data-based Mendelian Randomization (SMR) was used to further confirm colocalization (*Zhu et al., 2016*). The diverse ancestry sample genotypes in this study were used as the reference population as input to SMR.

## eQTL enrichment analysis

Enrichment of local-miRNA-eQTLs within functionally annotated genomic regions was done using GARFIELD v2 (*Iotchkova et al., 2019*) in order to control for the distance to transcription start sites, LD, minor allele frequency (MAF) of the tested variants, and the number of effective tests across multiple annotations. Functional annotations were derived from the Roadmap Epigenomics Project (*Kundaje et al., 2015*) for male and female fetal brain (E081 and E082), using the ChromHMM Core 15-state model (*Ernst and Kellis, 2012*). MAF and LD for the variants were derived from the 212, mixed-ancestry samples in this study using PLINK v1.9. Minimum local-miRNA-eQTL p-values were

used in cases where multiple association tests to different miRNAs were performed at a given variant. Only p-values surviving the stringent significance threshold were used for *Figure 2*, while other thresholds, including the relaxed threshold, can be seen in *Figure 2—figure supplement 1*.

## Comparison to blood miRNA-eQTLs

To assess the cell-type specificity of miRNA-eQTLs we calculated the $\pi_1$ statistic (*Storey and Tibshirani, 2003*). Blood miRNA-eQTLs were first defined as the emiR-variant pair with the lowest p-value for the 76 emiRs found in the blood miRNA-eQTL analysis (*Huan et al., 2015*). Of the 76 emiRs, 52 of these miRNA were expressed in brain tissue (at least 10 counts in at least 10 samples). At the 52 blood miRNA-eQTLs, nominal p-values from brain miRNA-eQTL association analysis were used to compute the $\pi_0$ value using the qvalue() function in the qvalue package (*Storey et al., 2021*). The $\pi_1$ statistic was then defined as $1 - \pi_0$. To estimate the standard error, we did 100 bootstrap samplings and computed a 95% confidence interval for each $\pi_1$ statistic. An analogous calculation was done using mRNA-eQTLs from an overlapping cortical tissue dataset (*Walker et al., 2019*) and whole blood mRNA-eQTLs reported by GTEx (*GTEx Consortium, Laboratory et al., 2017*).

## Lentiviral vector cloning and virus production

To create the miR-4707 expression vector, the tetracycline-inducible, lentiviral expression vector pTRIPZ (ThermoFisher RHS4750) was modified by replacing the sequence for red fluorescent protein (RFP) with the sequence for enhanced green fluorescent protein (EGFP) between the AgeI and ClaI restriction enzyme cut sites. The stem-loop sequence for hsa-mir-4707 (miRbase release 22) was inserted into the multiple cloning site of pTRIPZ-EGFP using XhoI and EcoRI restriction enzymes.

pTRIPZ-mir-4707-EGFP lentivirus was produced in HEK293T cells. HEK293T cells were obtained from ATCC (CRL-11268), identity verified by STR profiling, and tested negative for mycoplasma contamination. HEK293T cells were cultured to 90–95% confluency in DMEM (Life Technologies 11995081) supplemented with 10% FBS (Sigma-Aldrich F2442) and 1 x Antibiotic-Antimycotic (Life Technologies 15240096) in 10 cm tissue culture-treated plates. Cells were triple transfected with 10 µg transfer plasmid, 7.5 µg PAX2 (Addgene plasmid #12260), and 2.5 µg pMD2.G (Addgene plasmid #12259) using FUGENE HD (Promega E2311). After 24 hr, media was replaced with 12 mL of 1 x proliferation base media without the growth factors EGF, FGF, LIF, and PDGF (see progenitor cell culture protocol below). At 24 hr post media change, culture supernatant was filtered through a 0.45 µm syringe filter, aliquoted, and stored at –80 °C in single-use aliquots. Lentivirus was titered using the qPCR Lentivirus Titration Kit (Applied Biological Materials LV900).

## Primary Human Neural Progenitor Cell Culture

Primary human Neural Progenitor Cells (phNPCs) were derived from developing cortical tissue obtained from the UCLA Gene and Cell Therapy Core (see Methods: Tissue Procurement) as previously described (*Aygün et al., 2021*; *Liang et al., 2021*; *Stein et al., 2014*). Primary cell lines were determined to be free of mycoplasma contamination by RNA-sequencing, and genotypes were verified by RNA-sequencing using VerifyBamID (see Methods: Sample Quality Control). Two donor phNPC lines were used: Donor 54 (D54, gestation week 15.5, male, genotype G/G at rs4981455) and Donor 88 (D88, gestation week 14, male, genotype A/A at rs4981455). PhNPCs were grown on tissue culture treated plates that were coated with Growth Factor Reduced Matrigel (Corning 354230) at 50 µg/mL in 1xPBS at 37 °C for 1 hr. To maintain phNPCs in a proliferative state, they were cultured in 1 x proliferation media consisting of Neurobasal A (Life Technologies 10888–022) with 100 µg/ml primocin (Invivogen, ant-pm-2), 10% BIT 9500 (Stemcell Technologies 09500), 1 x glutamax (Life Technologies 35050061), 1 µg/ml heparin (Sigma-Aldrich, H3393-10KU), 20 ng/ml EGF (Life Technologies PHG0313), 20 ng/ml FGF (Life Technologies PHG0023), 2 ng/ml LIF (Life Technologies PHC9481), and 10 ng/ml PDGF (Life Technologies PHG1034). PhNPCs lines were split once per week using 0.05% Trypsin-EDTA (Life Technologies 25300062) into 1 x proliferation media. Every two or three days, half of the culture media was replaced with 2 x proliferation media: Neurobasal A with 100 µg/ml primocin, 10% BIT 9500, 1 x glutamax, 1 µg/ml heparin, 40 ng/ml EGF, 40 ng/ml FGF, 4 ng/ml LIF, and 20 ng/ml PDGF.

For phNPC proliferation experiments (*Figure 6—figure supplement 1*), cells were plated at $4 \times 10^5$ and $2 \times 10^4$ cells/well in Matrigel-coated 6-well (for RNA extractions: Corning 3516) and 96-well

(for immunocytochemistry: Corning 3598) plates respectively with 1 x proliferation media. At 24 hrs post plating, cells were transduced with pTRIPZ-mir-4707-EGFP or control (pTRIPZ-EGFP) at 20 IU/cell in 1 x proliferation media with 1 μg/ml doxycycline to express the miRNA (Sigma-Aldrich D9891). Media was changed at 24 hr post transduction with 1 x proliferation media with doxycycline, and plates were fed every 2 days by changing half of the culture media with 2 x proliferation media with 2 μg/ml doxycycline. At 8 days post transduction, six-well plates were used for RNA extraction, and 96-well plates were EdU labeled and fixed for immunofluorescence staining.

For differentiation experiments (*Figure 6*), D54 phNPCs were first transduced with pTRIPZ-mir-4707-EGFP or pTRIPZ-EGFP and stable lines were created to express mir-4707 (or control) under the control of doxycycline. phNPCs were plated in Matrigel-coated 10 cm dishes at $2.3 \times 10^6$ cells/dish in 1 x proliferation media. At 24 hr post plating, cells were transduced with pTRIPZ-mir-4707-EGFP or pTRIPZ-EGFP at 5 IU/cell. At 48 hr post transduction, puromycin (FisherScientific A1113803) was added to the culture media at a final concentration of 400 ng/mL to select for cells expressing the viral construct. A control plate of phNPCs that were non-transduced were also fed with puromycin containing media to ensure non-transduced cells would not survive. After 1 week in puromycin containing media, all non-transduced cells had died. Cells were transduced a second time as above but with 10 IU/cell in 10 cm dishes with $4 \times 10^6$ cells/dish coated in Matrigel. Stable lines were expanded in proliferation media (without puromycin) until ready to be plated for the differentiation experiments.

Stable lines of phNPCs containing pTRIPZ-mir-4707-EGFP or pTRIPZ-control were plated at $4.5 \times 10^5$ and $1.5 \times 10^4$ cells/well in Matrigel-coated 6-well and 96-well plates respectively in 1 x differentiation media: Neurobasal A (Life Technologies 10888–022) with 1 x Antibiotic-Antimycotic (Life Technologies 15240–062), 2% B27 (Life Technologies 17504–044), 1 x glutamax (Life Technologies 35050061), 10 ng/ml NT-3 (Life Technologies PHC7036) and 10 ng/ml BDNF (Life Technologies PHC7074). Every 2 or 3 days, half of the culture media was replaced with 2 x differentiation media: Neurobasal A with 100 μg/ml primocin, 2% B27, 1 x glutamax, 20 ng/ml NT-3 and 20 ng/ml BDNF. To turn on expression of mir-4707, doxycycline was added at 1 μg/ml on the day of plating to 1 x differentiation media. Doxycycline was maintained by addition of 2 μg/ml in 2 x differentiation media. At 1 and 2 weeks post differentiation, six-well plates were used for RNA extraction, and 96-well plates were fixed for immunocytochemistry and imaging.

## RNA extraction and qPCR for miRNA and mRNA expression

RNA was extracted from six-well plates using miRNeasy Mini Kits (QIAGEN 217004) with the inclusion of an on-column DNase digestion (QIAGEN 79254). Following elution with RNase-Free water, RNA concentration and quality was assessed with a NanoDrop ND-1000 Spectrophotometer. For cDNA synthesis of mRNA, iScript cDNA Synthesis Kits (Bio-Rad 1708891) were used with an input of 200 ng of total RNA. For cDNA synthesis of miRNA, TaqMan Advanced miRNA cDNA Synthesis Kits (ThermoFisher Scientific A28007) were used with 10 ng of total RNA input. Delta cycle threshold values (ΔCt) were calculated using the housekeeping gene EIF4A2, and fold change was calculated using control samples:

$$\Delta Ct = Ct_{target\ gene} - Ct_{housekeeping\ gene}$$

$$\Delta\Delta Ct = \Delta Ct_{sample} - \Delta Ct_{control}$$

$$Fold\ Change = 2^{-\Delta\Delta Ct}$$

To assay expression of mRNA, SsoAdvanced Universal SYBR Green Supermix (Bio-Rad 1725271) was used in 10 μL reactions on a 385-well plate. Reactions contained 2 μL of template cDNA (iScript reaction diluted 1:5 with water) and 500 nM of each forward and reverse primer (*Supplementary file 5*). Primers were chosen from the PrimerBank database (*Wang et al., 2012*). Reactions were placed in a QuantStudio 5 (Applied Biosystems) thermocycler and cycled for 40 cycles according to the SsoAdvanced protocol. To assay expression of miRNA, TaqMan Advanced miRNA Assays (ThermoFisher A25576) were used with probes against hsa-miR-361–5p as a housekeeping control and hsa-miR-4707–3p (Assay ID: 478056 and 479946). Reactions consisted of 2.5 μL of template cDNA (miRNA cDNA synthesis reaction diluted 1:10 with water), 0.5 μL of prob, and 5 μL of Taqman Fast Advanced Master Mix (ThermoFisher 4444557) in a 10 μL reaction. Reactions were placed in a QuantStudio 5 thermocycler and cycled for 45 cycles according to the TaqMan Advanced miRNA Assay protocol.

Delta cycle threshold values were calculated using hsa-miR-361–5p expression as the housekeeping gene, and fold change was calculated using control samples with the above equations. Technical replicates were defined as independent wells on a six-well plate. Technical replicates ranged from three to six; see figure legends for the number of technical replicates in each experiment. Significant differences were defined as a p-value <0.05 from a two-sided t-test.

## Microarray assay, PCA, and differential mRNA expression analysis

Extracted RNA from six wells of a six-well plate each of D54 pTRIPZ-mir-4707-EGFP or pTRIPZ-control from both 1 and 2 weeks, totaling 24 samples, was assayed for mRNA expression. The Clariom S Pico Assay (ThermoFisher 902963) microarray was used to quantify sample transcriptomes. Raw expression was normalized using the Robust Multichip Average (RMA) algorithm implemented in the R package, oligo (*Carvalho and Irizarry, 2010*). PCA analysis revealed cDNA yield as a technical batch effect, and the presence of two sample outliers (*Figure 6—figure supplement 2*). After removal of the batch effect and outliers, the 22 remaining samples segregated by expression vector and time point by PCA. Differential expression was done using limma (*Ritchie et al., 2015*), controlling for cDNA yield, and contrasting expression (miR-4707 vs control) in week 1 and week 2, or contrasting time point (week 1 vs week 2) in miR-4707 and control samples. Genes were considered significantly differentially expressed at a Benjamini-Hochberg adjusted p-value <0.05 (false discovery rate (FDR)<5%). Differentially expressed genes were used as input to a gene ontology analysis using all expressed genes as the background set of genes. For the gene ontology analysis in *Figure 6C*, all differentially expressed genes were used as input. For the gene ontology analysis in *Figure 6—figure supplement 2E and F*, only genes upregulated in week 1 or only genes upregulated in week 2, respectively, were used as input. Biological process ontologies were found to be significantly enriched by the 'elim' method and fisher's exact statistical test as implemented in the topGO R package (*A A, J R, 2022*).

## EdU assay, immunofluorescence labeling and imaging

To measure proliferating phNPCs, we measured incorporation of 5-ethynyl-2'-deoxyuridine (EdU) into newly synthesized DNA using a Click-iT EdU Imaging Kit with Alexa Fluor 647 (Invitrogen C10340). At 8 days post transduction, phNPCs in 1 x proliferation media in 96-well plates were labeled with the addition of 10 µM EdU at 37 °C for 2 hr. After the 2 hr incubation, cells were immediately fixed with 3.7% Paraformaldehyde (PFA) in PBS (FisherScientific 50-980-487) for 15 min. Fixed cells were then permeabilized with 0.5% Triton X-100 (Sigma-Aldrich T8787) for 20 min. Staining with Alexa Fluor 647 was conducted using the Click-iT EdU Imaging Kit protocol.

For proliferation experiments, fixed and permeabilized phNPCs in 96-well plates were blocked with 10% goat serum (MP Biomedicals 0219135680) in PBST: 0.02% Tween-20 (FisherScientific BP337500) in 1 x PBS, for 1 hr at room temperature. Cells were labeled with primary antibody in 3% goat serum/PBST at 4 °C overnight using 1:500 chicken-anti-GFP (FisherScientific AB16901). After overnight incubation, plates were washed three times for 5 min each with PBST. Secondary antibody labeling was done at 1:1000 dilution in 3% goat serum/PBST at room temperature for 1 hr using goat-anti-chicken-AF488 (ThermoFisher A11011). After 1 hr incubation, a 1:1000 dilution of DAPI (ThermoFisher 62248) in PBST was added to the secondary antibody solution for 10 min at room temperature. Plates were then washed three times for 5 min each with PBST. Plates were stored at 4 °C in 0.04% sodium azide (Sigma-Aldrich S2002) in PBS until imaging.

For differentiation experiments, phNPCs were fixed and permeabilized as above in 96-well plates and blocked with 10% goat serum (MP Biomedicals 0219135680) in PBST: 0.02% Tween-20 (FisherScientific BP337500) in 1 x PBS, for 1 hr at room temperature. Cells were labeled with primary antibody in 3% goat serum/PBST at 4 °C overnight using 1:500 chicken-anti-GFP (FisherScientific AB16901), 1:500 rabbit-anti-NESTIN (Millipore ABD69), and 1:1000 mouse-anti-TUJ1 (BioLegend 801202). After overnight incubation, plates were washed three times for 5 min each with PBST. Secondary antibody labeling was done at 1:1000 dilution in 3% goat serum/PBST at room temperature for 1 hr using goat-anti-chicken-AF488 (ThermoFisher A11011), goat-anti-rabbit-AF568 (ThermoFisher A11036), and goat-anti-mouse-AF750 (ThermoFisher A21037). After 1 hr incubation, a 1:1000 dilution of DAPI (ThermoFisher 62248) in PBST was added to the secondary antibody solution for 10 min at room temperature. Plates were then washed three times for 5 min each with PBST. Plates were stored at 4 °C in 0.04% sodium azide (Sigma-Aldrich S2002) in PBS until imaging.

Plates were imaged using a Nikon Eclipse Ti2 microscope set up for high-content image acquisition. For proliferation experiments, each well was imaged with 4 non-overlapping fields of view using ×10 magnification and a 0.3 numerical aperture using 3 filter sets: DAPI (ex: 325–375, em: 435–485), GFP (ex: 450–490, em: 500–550), and AF647 (ex: 625–655, em: 665–715). Image sets were fed through an automated CellProfiler (*McQuin et al., 2018*) pipeline to quantify the number of nuclei, GFP positive, and AF647 (EdU) positive nuclei in each image. Technical replicates were defined as independent wells of a 96-well plate. Replicates of 14 wells per condition. Significant differences were defined as a p-value <0.05 from a two-sided t-test. For differentiation experiments, each well was imaged with 4 non-overlapping fields of view using ×20 magnification and a 0.3 numerical aperture using 4 filter sets: DAPI (ex: 325–375, em: 435–485), GFP (ex: 450–490, em: 500–550), AF568 (ex: 520–600, em: 575–700), and AF750 (ex: 610–670, em: 625–775). Image sets were fed through an automated Cell-Profiler (*McQuin et al., 2018*) pipeline to quantify the number of nuclei and trace GFP-positive cells. Mean AF568 (NES) and AF750 (TUJ1) intensities were measured within GFP-positive cell outlines and normalized to the mean GFP intensity within each cell. The ratio of GFP normalized NES to GFP normalized TUJ1 was used to classify cells as neurons. The ratio cutoff was selected to enforce 20% of week 2 control cells as neurons. The number of neurons per well was then divided by the number of nuclei per well and reported in *Figure 6G*. Technical replicates were defined as independent wells of a 96-well plate. Replicates of 30 wells per condition. Significant differences were defined as a p-value <0.05 from a two-sided t-test.

## Acknowledgements

Tissue was collected from the UCLA CFAR (5P30 AI028697). RNA-seq libraries were sequenced by the UCLA Neuroscience Genomics Core. Microarray expression data was collected by the UNC Functional Genomics Core. The UNC Neuroscience microscopy core is supported by the NICHD (U54HD079124).

## Additional information

### Funding

| Funder | Grant reference number | Author |
| --- | --- | --- |
| National Institutes of Health | R01MH120125 | Jason L Stein |
| National Institute of General Medical Sciences | 5T32GM067553-13 | Michael J Lafferty |
| National Institutes of Health | R01MH118349 | Jason L Stein |
| National Institutes of Health | U54EB020403 | Jason L Stein |
| National Institutes of Health | R00MH102357 | Jason L Stein |

The funders had no role in study design, data collection and interpretation, or the decision to submit the work for publication.

### Author contributions

Michael J Lafferty, Data curation, Formal analysis, Validation, Investigation, Visualization, Methodology, Writing - original draft, Writing - review and editing, Designed validation experiments; performed validation experiments; performed bioinformatic analyses; Nil Aygün, Writing - review and editing, Performed the mRNA-e/sQTL analysis; supported bioinformatic analyses; Niyanta K Patel, Writing - review and editing, Performed validation experiments; supported bioinformatic analyses; Oleh Krupa, Writing - review and editing, Supported validation experiments; supported bioinformatic analyses; Dan Liang, Data curation, Writing - review and editing, Preprocessed sample genotype data; supported bioinformatic analyses; Justin M Wolter, Methodology, Writing - review and editing, Supported validation experiments; supported validation experiments; Daniel H Geschwind,

Conceptualization, Resources, Funding acquisition, Methodology, Writing - review and editing; Luis de la Torre-Ubieta, Conceptualization, Supervision, Project administration, Writing - review and editing, Collected samples; designed validation experiments; Jason L Stein, Conceptualization, Resources, Supervision, Funding acquisition, Investigation, Methodology, Writing - original draft, Project administration, Writing - review and editing, Collected samples; designed validation experiments

## Author ORCIDs
Michael J Lafferty (iD) http://orcid.org/0000-0002-1000-0480
Daniel H Geschwind (iD) http://orcid.org/0000-0003-2896-3450
Jason L Stein (iD) http://orcid.org/0000-0003-4829-0513

## Ethics
Human fetal brain tissue was obtained from the UCLA Gene and Cell Therapy Core following institutional review board regulations. This study was declared Exempt by the UNC Institutional Review Board (16-0054).

## Decision letter and Author response
Decision letter https://doi.org/10.7554/eLife.79488.sa1
Author response https://doi.org/10.7554/eLife.79488.sa2

## Additional files

### Supplementary files
• Supplementary file 1. Expressed known and novel miRNAs. UNIQUE_NAME: combination of primary-miRNA and mature-miRNA name to create a unique identifier for miRNAs with the same sequence that originate from multiple genomic loci. ID: miRBase ID, NAME for non-miRBase miRNAs. ALIAS: miRBase alias, NAME for non-miRBase miRNAs. NAME: miRBase name, for non-miRBase miRNAs the identifier given in the respective publications or identifier given by miRge2.0 or miRDeep2 packages. DERIVES_FROM: ID of primary-miRNA. DERIVES_FROM_NAME: NAME of primary-miRNA. SOURCE: miRBase_v22 for known miRNAs cataloged in miRBase release 22. Friedlander2014 or Nowakowski2018 for novel miRNAs reported in those respective publications. miRge or miRDeep2 for novel miRNAs discovered in this study. TYPE: miRNA for mature miRNAs in miRBase, miRNA_putative_mature or miRNA_putative_star for novel miRNAs. SCORE: NA for miRBase and Nowakowski2018 miRNAs. Friedlander2014 miRNAs reported a confidence score 1–4. miRge quality scores 0–1. miRDeep2 scores ≥ 0. CHR: chromosome where the mature miRNA is located. START_hg38: start base-pair position for mature miRNA using hg38. END_hg38: end base-pair position for the mature miRNA using hg38. WIDTH: base-pair width of the mature miRNA. STRAND: genomic strand of the mature miRNA. SEQUENCE: miRNA sequence. MEAN_VST_EXPRESSION: mean variance-stabilizing transformation expression.

• Supplementary file 2. Differentially expressed known and novel miRNAs. BASE_MEAN: mean expression as reported by DESeq2. LOG2_FOLD_CHANGE: log2 transformed fold-change as reported by DESeq2 on the treatment variable, gestation week. Positive values indicate enrichment within late gestation week tissues. Negative values indicate enrichment within early gestation week tissue. LFC_SE: standard error on fold change as reported by DESeq2. PVALUE: p-value as reported by DESeq2. PADJ: Benjamini-Hochberg adjusted p-value as reported by DESeq2. SIGNIFICANT: logical, TRUE if PADJ is below 0.1. NAME: same as in *Supplementary file 1*. ID: same as in *Supplementary file 1*. ALIAS: same as in *Supplementary file 1*. DERIVES_FROM: same as in *Supplementary file 1*. SOURCE: same as in *Supplementary file 1*. TYPE: same as in *Supplementary file 1*. SCORE: same as in *Supplementary file 1*. SEQUENCE: same as in *Supplementary file 1*.

• Supplementary file 3. Mid-gestation cortical tissue miRNA-eQTLs. eQTL: unique eQTL identifier, combination of emiR and eSNP. emiR: same as UNIQUE_NAME in *Supplementary file 1*. eSNP: unique variant identifier, combination of chromosome, base-pair position, and variants. BETA: eQTL association effect size after fitting to the linear mixed model using EMMAX. P: nominal p-value on linear mixed model fitting using EMMAX. DEGREE: degree to which the eQTL is conditionally-independent. SNP_CHR: variant chromosome. SNP_BP_hg38: variant base-pair position using hg38. EFFECT_ALLELE: effect allele used in the linear mixed model association by EMMAX. REF: reference allele using hg38. ALT: alternate allele using hg38. ALT_CTS: alternate allele counts. Summed allelic dosage across all samples in the analysis. OBS_CT: total allele counts. Number of samples x2. A1:

A1 allele as defined by plink1.9, usually minor allele. A2: A2 allele as defined by plink1.9, usually major allele. A1_HOM_COUNT: number of homozygous A1 samples. HET_COUNT: number of heterozygous samples. A2_HOM_COUNT: number of homozygous A2 samples. miR_CHR: miRNA chromosome. miR_START_hg38: miRNA start position using hg38. miR_END_hg38: miRNA end position using hg38. miR_WIDTH: miRNA width in base pairs. miR_STRAND: miRNA genomic strand. SOURCE: same as in *Supplementary file 1*. TYPE: same as in *Supplementary file 1*. ID: same as in *Supplementary file 1*. ALIAS: same as in *Supplementary file 1*. NAME: same as in *Supplementary file 1*. DERIVES_FROM: same as in *Supplementary file 1*. DERIVES_FROM_NAME: same as in *Supplementary file 1*. SEQUENCE: same as in *Supplementary file 1*. SIGNIFICANCE: label for significance threshold used to define significant eQTLs, eigenMT_fdr5percent is the stringent threshold and fdr5percent is the relaxed threshold, see Methods. NOM_P_VALUE_THRESHOLD: nominal p-value threshold used to define significant eQTLs.

• Supplementary file 4. Colocalizations. Sheet1: miRNA-eQTL/mRNA-eQTL colocalizations. Columns with.mirQTL suffix refer to the miRNA-eQTL analysis while.mQTL refer to the mRNA-eQTL analysis (*Aygün et al., 2021*; *Walker et al., 2019*). eQTL.mirQTL: unique miRNA-eQTL identifier, combination of emiR and eSNP. emiR.mirQTL: same as UNIQUE_NAME in *Supplementary file 1*. eSNP.mirQTL: unique variant identifier, combination of chromosome, base-pair position, and variants. BETA.mirQTL: eQTL association effect size after fitting to the linear mixed model using EMMAX. P.mirQTL: nominal p-value on linear mixed model fitting using EMMAX. DEGREE.mirQTL: degree to which the eQTL is conditionally-independent.SNP.CHR.mirQTL: variant chromosome. SNP.BP.hg38.mirQTL: variant base-pair position using hg38. EFFECT.ALLELE.mirQTL: effect allele used in the linear mixed model association by EMMAX. REF.mirQTL: reference allele using hg38. ALT.mirQTL: alternate allele using hg38. SIGNIFICANCE.mirQTL: label for significance threshold used to define significant eQTLs, eigenMT_fdr5percent is the stringent threshold and fdr5percent is the relaxed threshold, see Methods. SNP.mQTL: variant identifier. ENSG.mQTL: Ensembl gene ID. BETA.mQTL: mRNA-eQTL beta value. P.mQTL: mRNA-eQTL p-value. CHR.mQTL: variant chromosome. BP.mQTL: variant base-pair position on hg38. RANK.mQTL: conditional analysis rank. BETA.CONDITIONAL.mQTL: beta after conditioning. P.CONDITIONAL.mQTL: p-value after conditioning. ALLELE_MINOR.mQTL: minor allele in the mRNA-eQTL dataset. ALLELE_MAJOR_EFFECT.mQTL: effect allele. Major allele in the mRNA-eQTL dataset. eQTL.mQTL: unique mRNA-eQTL identifier, combination of SNP.mQTL and ENSG.mQTL. Sheet2: miRNA-eQTL/mRNA-sQTL colocalizations. Columns with.mirQTL suffix refer to the miRNA-eQTL analysis while.mQTL refer to the mRNA-sQTL analysis (*Aygün et al., 2021*; *Walker et al., 2019*). eQTL.mirQTL: see Sheet 1. emiR.mirQTL: see Sheet 1. eSNP.mirQTL: see Sheet 1. BETA.mirQTL: see Sheet 1. P.mirQTL: see Sheet 1. DEGREE.mirQTL: see Sheet 1. SNP.CHR.mirQTL: see Sheet 1. SNP.BP.hg38.mirQTL: see Sheet 1. EFFECT.ALLELE.mirQTL: see Sheet 1. REF.mirQTL: see Sheet 1. ALT.mirQTL: see Sheet 1. SIGNIFICANCE.mirQTL: see Sheet 1. snp.mQTL: unique variant identifier, combination of chromosome, base-pair position, and variants. intron.mQTL: unique intron identifier, combination of chromosome, base-pair start and end positions, and cluster identifier. beta.mQTL: mRNA-sQTL beta value. pvalue.mQTL: mRNA-sQTL p-value. chr.mQTL: sQTL chromosome. rank.mQTL: degree to which the sQTL is conditionally independent. cond.beta.mQTL: mRNA-sQTL beta value at conditional rank. cond.pval.mQTL: mRNA-sQTL p-value at conditional rank. clusterID.mQTL: unique cluster identifier. gene.mQTL: gene symbol. ensemblID.mQTL: Ensembl gene ID. transcripts.mQTL: Ensembl transcript ID. BP.mQTL: sQTL base-pair. rsid.mQTL: rsid of sQTL. Sheet3: miRNA-eQTL brain/blood colocalizations eQTL.mirQTL: see Sheet 1. emiR.mirQTL: see Sheet 1. eSNP.mirQTL: see Sheet 1. BETA.mirQTL: see Sheet 1. P.mirQTL: see Sheet 1. DEGREE.mirQTL: see Sheet 1. SNP.CHR.mirQTL: see Sheet 1. SNP.BP.hg38.mirQTL: see Sheet 1. EFFECT.ALLELE.mirQTL: see Sheet 1. REF.mirQTL: see Sheet 1. ALT.mirQTL: see Sheet 1. SIGNIFICANCE.mirQTL: see Sheet 1. NOM.P.VALUE.THRESH.mirQTL: see Sheet 1. snpID.bloodQTL: rsid for blood eQTL. Estimate.bloodQTL: blood eQTL beta. Pval.bloodQTL: blood eQTL p-value. hsa_miR_name.bloodQTL: miRBase miRNA name. effect.bloodQTL: effect variant for blood eQTL. noneffect.bloodQTL: non-effect variant for blood eQTL. Sheet4: GWAS data sources TRAIT: trait or disorder name. PMID: PubMed ID for published article associated with each dataset. DATA_LINK: link to data download site.

• Supplementary file 5. qPCR primers. GENE: gene name. NCBI_GENE_ID: NCBI gene ID. PRIMER_BANK_ID: primer bank ID. AMPLICON_SIZE: distance between forward and reverse primer on gene transcript. FORWARD_PRIMER: forward primer. REVERSE_PRIMER: reverse primer.

• Supplementary file 6. Differentially expressed genes in phNPCs. Sheet1: Expression in Week1: microarray differential gene expression associated with over-expression of miR-4707–3p at week 1. PROBEID: pd.clariom.s.human.ht probe IDs. SYMBOL: gene symbol. GENENAME: gene name. ENSEMBL: Ensembl gene ID. ENTREZID: Entrez gene ID. duplicate.annotation: TRUE if probe

ID is associated with more than one ensembl gene id. logFC: log-fold-change values as reported by limma. Positive values represent increased expression in control samples relative to mir-4707 samples. AveExpr: average normalized expression. t: absolute t-statistic as reported by limma. P.Value: raw p-values as reported by limma. adj.P.Val: Benjamini-Hochberg adjusted p-value as reported by limma. B: B-statistic as reported by limma. Sheet2: Expression in Week2: microarray differential gene expression associated with over-expression of miR-4707–3p at week 2. PROBEID: pd.clariom.s.human.ht probe IDs. SYMBOL: gene symbol. GENENAME: gene name. ENSEMBL: Ensembl gene ID. ENTREZID: Entrez gene ID. duplicate.annotation: TRUE if probe ID is associated with more than one ensembl gene id. logFC: log-fold-change values as reported by limma. Positive values represent increased expression in control samples relative to mir-4707 samples. AveExpr: average normalized expression. t: absolute t-statistic as reported by limma. P.Value: raw p-values as reported by limma. adj.P.Val: Benjamini-Hochberg adjusted p-value as reported by limma. B: B-statistic as reported by limma. Sheet3: Diff in Control: microarray differential gene expression associated with differentiation (week1 vs week2) in control samples. PROBEID: pd.clariom.s. human.ht probe IDs.SYMBOL: gene symbol. GENENAME: gene name. ENSEMBL: Ensembl gene ID. ENTREZID: Entrez gene ID. duplicate.annotation: TRUE if probe ID is associated with more than one ensembl gene id. logFC: log-fold-change values as reported by limma. Positive values represent increased expression in week 2 control samples relative to week 1 control samples. AveExpr: average normalized expression. t: absolute t-statistic as reported by limma. P.Value: raw p-values as reported by limma. adj.P.Val: Benjamini-Hochberg adjusted p-value as reported by limma. B: B-statistic as reported by limma. Sheet4: Diff in 4707: microarray differential gene expression associated with differentiation (week1 vs week2) in miR-4707–3p over-expression samples. PROBEID: pd.clariom.s.human.ht probe IDs. SYMBOL: gene symbol. GENENAME: gene name. ENSEMBL: Ensembl gene ID. ENTREZID: Entrez gene ID. duplicate.annotation: TRUE if probe ID is associated with more than one ensembl gene id. logFC: log-fold-change values as reported by limma. Positive values represent increased expression in week 2 mir-4707 samples relative to week 1 mir-4707 samples. AveExpr: average normalized expression. t: absolute t-statistic as reported by limma. P.Value: raw p-values as reported by limma. adj.P.Val: Benjamini-Hochberg adjusted p-value as reported by limma. B: B-statistic as reported by limma.

- MDAR checklist

## Data availability

Small RNA-sequencing data and sample genotypes will be available via dbGaP with study accession number phs003106.v1.p1. Total RNA-sequencing data can be found under the dbGaP study phs001900.v1.p1. Scripts used to reproduce the analyses presented here are available via GitHub code repository at https://github.com/mikelaff/mirna-eqtl-manuscript (copy archived at swh:1:rev:50507250189b0b21bfc2d70c81e0ebaea1be39c6).

The following dataset was generated:

| Author(s) | Year | Dataset title | Dataset URL | Database and Identifier |
|---|---|---|---|---|
| Lafferty MJ, Aygün N, Krupa O, Liang D, Wolter JM, Geschwind DH, de la Torre-Ubieta L, Stein JL | 2022 | Genetic effects on miRNA expression during mid-gestation neocortical development | https://www.ncbi.nlm. nih.gov/projects/gap/ cgi-bin/study.cgi? study_id=phs003106. v1.p1 | dbGaP, phs003106.v1.p1 |

The following previously published dataset was used:

| Author(s) | Year | Dataset title | Dataset URL | Database and Identifier |
|---|---|---|---|---|
| Walker RL, Ramaswami G, Hartl C, Mancuso N, Gandal MJ, de la Torre-Ubieta L, Pasaniuc B, Stein JL, Geschwind DH | 2020 | Genetic Control of Expression and Splicing in Developing Human Brain | https://www.ncbi.nlm. nih.gov/projects/gap/ cgi-bin/study.cgi? study_id=phs001900. v1.p1 | dbGaP, phs001900.v1.p1 |

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
