## [Editor Report]

Lafferty et al. study the regulation of microRNA (miRNA) levels in mid-gestation human neocortical tissues as a potential contributor to brain-related phenotypes. They performed miRNA expression profiling and correlated expression quantitative trait loci (eQTLs) that locally regulate miRNA genes, searching for potential overlap. They report colocalization at SNP rs4981455 which is an eQTL for miR-4707-3p and is also associated with global cortical surface area (GSA) and educational attainment phenotypes in GWAS, and demonstrate exogenously increased expression of miR-4707-3p in primary human neural progenitor cells increases the rate of proliferation. The reported results are interesting and important, particularly for the understanding of miRNA biology, and suggest long-suspected roles for miRNAs in neurogenesis and human brain disease.

---

## [Decision Letter]

**Decision letter after peer review:**

Thank you for submitting your article "MicroRNA-eQTLs in the developing human neocortex link miR-4707-3p expression to brain size" for consideration by *eLife*. Your article has been reviewed by 2 peer reviewers, and the evaluation has been overseen by a Reviewing Editor and Molly Przeworski as the Senior Editor. The following individual involved in the review of your submission has agreed to reveal their identity: Giordano Lippi (Reviewer #1).

Essential revisions:

1. Tone down the strong statements they make throughout the paper, e.g., "Here we highlight one example of how miRNA expression leads to differences in brain size and cognition…".

2. Avoid presenting a causal pathway as shown in Figure 6F (at least the last step regarding the GWAS traits).

3. Add a discussion of caveats and implications, i.e., what the findings mean and don't mean.

*Reviewer #1 (Recommendations for the authors):*

This is an excellent paper that will serve well the broad the miRNA and brain development community. A few concerns limit the enthusiasm (note: my expertise is in miRNA mechanisms of neuronal development, so I cannot comment on the quality/validity of the genomics approach, for which I hope appropriate reviewers have been selected):

1. Figure 5B shows that the A/A genotype at rs4981455 has a robust increase in miR-4707-3p levels compared to the G/G genotype. However, the results with the D54 (G/G) and D88 (A/A) lines are a bit confusing. First, why do you have a larger increase in miR-4707-3p expression in D54 (that if I understand well shouldn't have any miR-4707-3p expressed) than in D88 (Fig6B)? Second, why is overexpression of miR-4704-3p in D54 inducing a much smaller neurogenesis effect than endogenous expression in D88 (Fig6D)? Third, why is the over-expression of miR-4707-3p in D88 inducing a larger neurogenesis effect, considering that it was already expressed? These results are counterintuitive.

2. The authors conclude that depletion of progenitors is what causes a decrease in the cortical surface. I think this conclusion is not fully supported by the data. An increase in neurogenesis does not necessarily mean a decrease in progenitors. This needs to be tested experimentally.

*Reviewer #2 (Recommendations for the authors):*

- It'd be useful to explore/present additional features of miRNA-eQTLs, e.g., distribution of effect sizes across top eQTLs per gene (how much of variance they explain), distribution of cis-heritabilities across genes, number of genes affected per eQTL, etc.

- I found Figure 2A confusing. Specifically, a standard Manhattan plot in GWAS shows p-values for the association across loci with the same phenotype. But in Figure 2A p-values are technically for different expression phenotypes, for genes with variable cis-heritabilities. For example, what is plotted when one variant affects multiple genes? It may be desirable to consider an alternative presentation, though I don't deem it necessary.

- I think there are typos in the paragraph in lines 192-204, related to increasing/decreasing effect of A/G alleles.

- Related to Figure 5, it'd be useful to include locus plots for the other neurological and behavioral traits studied as a supplementary plot.

---

## [Author Response]

Essential revisions:1. Tone down the strong statements they make throughout the paper, e.g., "Here we highlight one example of how miRNA expression leads to differences in brain size and cognition…".

We have modified many of the strong statements made throughout the paper to mitigate misinterpretation of our findings. Some examples include:

(from the original abstract) “Integrating miRNA-eQTLs with existing GWAS yielded discovery of a miRNA modulating developmental fate decisions that alter human brain size.”

(updated abstract) “Integrating miRNA-eQTLs with existing GWAS yielded evidence of a miRNA that may influence human brain size and function via modulation of neocortical brain development.”

(from the original discussion) “Here we highlight one example of how miRNA expression leads to differences in brain size and cognition through altered neurogenesis during cortical development.”

(updated discussion) "In this study we highlight one example of how individual differences in miRNA expression during fetal cortical development may lead to differences in brain size and cognition through altered neurogenesis.”

We hope that these modified statements maintain the significance of our findings while conceding that we do not have sufficient evidence to prove a causal link between miR-4707-3p expression during cortical development and adult brain phenotypes.

2. Avoid presenting a causal pathway as shown in Figure 6F (at least the last step regarding the GWAS traits).

Figure 6F (now presented in Figure 6H) has been updated to show an uncertain biological mechanism between rs4981455, miR-4707-3p expression, and adult GWAS traits.

3. Add a discussion of caveats and implications, i.e., what the findings mean and don't mean.

We have added a paragraph to the discussion on the caveats and implications of our findings.

Reviewer #1 (Recommendations for the authors):This is an excellent paper that will serve well the broad the miRNA and brain development community. A few concerns limit the enthusiasm (note: my expertise is in miRNA mechanisms of neuronal development, so I cannot comment on the quality/validity of the genomics approach, for which I hope appropriate reviewers have been selected):1. Figure 5B shows that the A/A genotype at rs4981455 has a robust increase in miR-4707-3p levels compared to the G/G genotype. However, the results with the D54 (G/G) and D88 (A/A) lines are a bit confusing. First, why do you have a larger increase in miR-4707-3p expression in D54 (that if I understand well shouldn't have any miR-4707-3p expressed) than in D88 (Fig6B)? Second, why is overexpression of miR-4704-3p in D54 inducing a much smaller neurogenesis effect than endogenous expression in D88 (Fig6D)? Third, why is the over-expression of miR-4707-3p in D88 inducing a larger neurogenesis effect, considering that it was already expressed? These results are counterintuitive.

We thank the reviewer for their positive evaluation of our work and helpful comments. We understand the confusion that rs4981455 genotypes of D54 and D88 phNPC cell lines may cause. Therefore we have added additional text describing the experiments in Figure 6. First, we now explicitly state the genotype at rs4981455 of both D54 and D88 cell lines in the text:

“We transduced phNPCs from two genetically distinct donors: Donor 54 (D54, genotype G/G at rs4981455) and Donor 88 (D88, genotype A/A at rs4981455) that were cultured in media…”

Second, we have also added a panel to Figure 6—figure supplement 1 which shows the qPCR mean CT values for both control samples (endogenous expression) and mir-4707 samples (endogenous plus exogenous expression).

Finally, we have added text describing the endogenous expression of miR-4707-3p in D54 and D88:

“PhNPCs derived from D88 (homozygous for the miR-4707-3p increasing allele) endogenously expressed miR-4707-3p at a greater level than phNPCs derived from D54 (homozygous for the miR-4707-3p decreasing allele), as expected given their genotypes (Figure 6—figure supplement 1C).”

The large increase in miR-4707-3p expression shown in D54 fold-change is most likely a result of endogenous D54 miR-4707-3p expression being very low. We hope that including the mean CT values in the figure supplement will clear up confusion surrounding endogenous expression of miR-4707-3p in the phNPC lines.

Lastly, the experiment presented in Figure 6 (now Figure 6—figure supplement 1) using proliferating phNPCs left many questions unanswered as to the direction in which miR-4707-3p over-expression was guiding the phNPC lines. We saw increased proliferative as well as increased neurogenic markers, making us unable to conclude if miR-4707-3p increased or decreased neurogenesis. This led us to an additional experiment using differentiating phNPCs (now presented in Figure 6).

2. The authors conclude that depletion of progenitors is what causes a decrease in the cortical surface. I think this conclusion is not fully supported by the data. An increase in neurogenesis does not necessarily mean a decrease in progenitors. This needs to be tested experimentally.

We agree that the experiment in the initial submission using proliferating phNPCs was not able to fully support a biological mechanism between increased miR-4707-3p expression and decreased cortical surface area. In that experiment, we saw both increased proliferative and neurogenic gene markers in response to increased miR-4707-3p expression. We hypothesized that miR-3707-3p expressing phNPCs are exiting the cell cycle and differentiating into neurons at a greater rate than control phNPCs.

To test this hypothesis experimentally, we designed a second experiment with phNPCs in which the media growth factors maintaining a proliferative state are removed and the cells are allowed to differentiate. This new experiment is now the focus of Figure 6, and the previous experiment using proliferating phNPCs has been moved to Figure 6—figure supplement 1. In the updated experiment with differentiating phNPCs we measured the overall transcriptomic changes induced by miR-4707-3p over-expression using a microarray assay. We also measured the proportion of neurons and progenitors in culture using immunocytochemistry.

The following paragraphs have been added to the Results section, “miR-4707-3p modulates target gene expression and proliferation in phNPCs” which describes these experimental findings:

“We next sought to better characterize miR-4707-3p’s effect on neurogenesis by modulating its expression in phNPCs grown in media where proliferative growth factors were removed, allowing differentiation (25,65). […] Though the decrease in neurogenesis at week one caused by miR-4707-3p is not consistent with the mechanism leading to cortical surface area decrease proposed by the radial unit hypothesis, our experiments nevertheless confirm that miR-4707-3p increases phNPC proliferation and modulates synaptic and neuronal maturation genes.”

As shown in Figure 6, and discussed in the text, we were not able to show definitively that miR-4707-3p over-expression results in decreased progenitors or increased neurons. However, the transcriptomic changes and decreased expression of predicted miR-4707-3p targets reveals the effect of miR-4707-3p over-expression may influence neuronal maturation or synaptogenesis. These phenotypes also point to a biological mechanism that may explain the genomic colocalization with educational attainment.

Since we are not able to test the effect of miR-4707-3p expression on downstream brain phenotypes, the biological mechanism connecting these phenotypes is not consistent with the predictions of the radial unit hypothesis. However, we believe the experimental evidence presented here will help drive future hypotheses linking genetic control of miRNA expression with genetic associations to brain traits. We have updated Figure 6H to show our uncertainty with the biological link between miR-4707-3p expression and adult brain traits and is discussed in greater detail below.

Reviewer #2 (Recommendations for the authors):- It'd be useful to explore/present additional features of miRNA-eQTLs, e.g., distribution of effect sizes across top eQTLs per gene (how much of variance they explain), distribution of cis-heritabilities across genes, number of genes affected per eQTL, etc.

We have added Figure 2—figure supplement 1 which now shows the number of eGenes (emiRs) per eSNP and the distribution of percent of variance explained by each miRNA-eQTL:

Because most emiRs are associated with only one eSNP, we did not estimate the distribution of cis-heritabilities across these miRNAs.

- I found Figure 2A confusing. Specifically, a standard Manhattan plot in GWAS shows p-values for the association across loci with the same phenotype. But in Figure 2A p-values are technically for different expression phenotypes, for genes with variable cis-heritabilities. For example, what is plotted when one variant affects multiple genes? It may be desirable to consider an alternative presentation, though I don't deem it necessary.

We understand how a Manhattan plot of local-miRNA-eQTLs across the genome may be confused with Manhattan plots typically representing GWAS p-values. However, we wished to show the breadth of the eQTLs discovered in this study and represent them in an eye-catching summary figure. We have added language to the figure legend clarifying how p-values are represented when variants affect multiple miRNAs:

“Variants tested for association to more than one expressed miRNA are represented by independent points for each association p-value.”

- I think there are typos in the paragraph in lines 192-204, related to increasing/decreasing effect of A/G alleles.

We believe the description of the effect direction of the A and G alleles in this paragraph is correct. However, we have added to this paragraph to decrease confusion about the effect of each allele on splicing and miRNA expression:

“Among these splice sites, we observed an association between genotypes (rs7911488 alleles A/G) and splice site utilization at SpliceA and SpliceD. This same variant was associated with expression of miR-1307-5p (Figure 3D). The G allele was associated with an increased PSI of SpliceA and SpliceD into the mRNA, while these same samples showed a decreased expression of miR-1307-5p.”

- Related to Figure 5, it'd be useful to include locus plots for the other neurological and behavioral traits studied as a supplementary plot.

The miRNA-eQTL for miR-4707-3p expression shown in Figure 5 colocalized with only one neurological or behavior trait, educational attainment, from a total of 21 traits we tested for colocalization. Therefore, we felt that adding additional traits, which don’t show a genomic colocalization at this locus would confuse the reader. The miR-4707-3p eQTL colocalization to brain size phenotypes (head size and intracranial volume) were not plotted in Figure 5 because the summary statistics for these studies have not yet been made public.